# Advances in the Molecular Landscape of Lung Cancer Brain Metastasis

**DOI:** 10.3390/cancers15030722

**Published:** 2023-01-24

**Authors:** Vanessa G. P. Souza, Rachel Paes de Araújo, Mariana R. Santesso, Ana Laura Seneda, Iael W. Minutentag, Tainara Francini Felix, Pedro Tadao Hamamoto Filho, Michelle E. Pewarchuk, Liam J. Brockley, Fábio A. Marchi, Wan L. Lam, Sandra A. Drigo, Patricia P. Reis

**Affiliations:** 1Molecular Oncology Laboratory, Experimental Research Unit, Faculty of Medicine, São Paulo State University (UNESP), Botucatu 18618-687, Brazil; 2British Columbia Cancer Research Institute, Vancouver, BC V5Z 1L3, Canada; 3Department of Neurology, Psychology and Psychiatry, Faculty of Medicine, São Paulo State University (UNESP), Botucatu 18618-687, Brazil; 4Faculty of Medicine, University of São Paulo, São Paulo 01246-903, Brazil; 5Department of Surgery and Orthopedics, Faculty of Medicine, São Paulo State University (UNESP), Botucatu 18618-687, Brazil

**Keywords:** brain metastasis, lung cancer, microenvironment, molecular mechanisms, coding and non-coding RNAs, therapeutic strategies

## Abstract

**Simple Summary:**

Patients with lung cancer have high rates of brain metastasis (BM). Despite available therapies, patient prognosis is poor. Studies have shown genetic alterations associated with the metastatic spread of lung cancer cells. However, the precise mechanisms governing BM are still unclear. In this review, we comprehensively describe the major steps of metastatic spread of lung cancer to the brain, addressing the influence of the tumor microenvironment and the molecular determinants of progression. Furthermore, we highlight the advances in the molecular diagnostics of BM by liquid biopsies and discuss novel treatment strategies.

**Abstract:**

Lung cancer is one of the most frequent tumors that metastasize to the brain. Brain metastasis (BM) is common in advanced cases, being the major cause of patient morbidity and mortality. BMs are thought to arise via the seeding of circulating tumor cells into the brain microvasculature. In brain tissue, the interaction with immune cells promotes a microenvironment favorable to the growth of cancer cells. Despite multimodal treatments and advances in systemic therapies, lung cancer patients still have poor prognoses. Therefore, there is an urgent need to identify the molecular drivers of BM and clinically applicable biomarkers in order to improve disease outcomes and patient survival. The goal of this review is to summarize the current state of knowledge on the mechanisms of the metastatic spread of lung cancer to the brain and how the metastatic spread is influenced by the brain microenvironment, and to elucidate the molecular determinants of brain metastasis regarding the role of genomic and transcriptomic changes, including coding and non-coding RNAs. We also present an overview of the current therapeutics and novel treatment strategies for patients diagnosed with BM from NSCLC.

## 1. Introduction

Non-Small Cell Lung Cancer (NSCLC) accounts for the majority (85%) of lung cancer cases, with adenocarcinoma and squamous cell carcinoma being the most common histological subtypes [1]. Tobacco smoking, air pollution, and exposure to radiation and occupational carcinogens are among the most common risk factors [2]. Increased incidence of lung cancer has been observed in never-smokers and younger individuals and is frequently associated with the adenocarcinoma subtype. It is expected that by 2040, the worldwide incidence of lung cancer will increase from 2 to over 3 million cases per year, and the number of annual deaths will rise from 1.8 to over 2.9 million [3]. A limited proportion of lung cancer patients undergo surgery as the primary treatment since most patients (~75%) present locally advanced or distant metastatic disease at diagnosis and are not eligible for curative surgical treatment. Since current treatment strategies are focused on treating late-stage disease, patient prognosis remains dismal, with high mortality rates.

Metastatic disease is one of the main causes of patient death. Therefore, uncovering the mechanisms underlying metastasis is pivotal for improving therapeutic strategies and patient survival [4].

Previous treatment of advanced lung cancer was limited to cytotoxic chemotherapy, but the identification of oncogenic driver mutations in NSCLC has dramatically changed the therapeutic approaches in the past decades. Targeted therapy and immunotherapy have considerably improved survival in selected patients [1]. Large-scale genomic studies have enabled the identification of activating driver mutations associated with primary lung cancer. These findings contributed to advances in therapeutics with the development of tyrosine kinase inhibitors (TKIs), which led to improvements in patient survival. However, these therapies benefit a fraction of patients with lung adenocarcinoma harboring driver mutations. Mutations in genes such as the Epidermal Growth Factor Receptor (*EGFR*), Anaplastic Lymphoma Kinase (*ALK*), ROS1 Proto-Oncogene Tyrosine Kinase Receptor (*ROS1*), and Serine/Threonine-Protein Kinase BRAF (*BRAF*) are therapeutic targets in lung adenocarcinoma, and novel mutations may be introduced as targeted therapies [5]. Other activating mutations occur in oncogenes such as *KRAS* and are associated with worse prognosis, with no approved drugs able to efficiently inhibit *KRAS* activation [5,6,7] until the recent development of novel inhibitors such as sotorasib, shown to efficiently target the KRASp.G12C mutation in advanced solid tumors [8]. Another KRAS inhibitor, adagrasib, is under investigation to treat patients with progressive metastatic lung cancer [9]. In a phase 2 clinical trial, sotorasib, which specifically and irreversibly inhibits the KRASp.G12C mutation, was tested in a cohort of 126 NSCLC patients, with the majority having previously received systemic platinum-based chemotherapy combined with immunotherapy based on PD-1 or PD-L1 immune checkpoint inhibitors (ICIs). Results showed a complete response in 4/126 patients (3.2%) and a partial response in 42/126 patients (33.9%) with a median duration of response of 11.1 months. These data showed a clinical response for relapsed advanced *KRAS*-mutated NSCLC with disease control obtained in >80% of patients. Median progression-free survival and overall survival were 6.8 months and 12.5 months, respectively [10]. Novel treatments targeting *KRAS*-mutated tumors are promising; however, patient prognosis remains poor with modest progression-free and overall survival rates, likely due to disease heterogeneity. Recent reviews on targeted therapies with KRAS inhibitors including a combination with immunotherapies are available in [11,12,13].

Mutations in *EGFR* occur in 15–40% of adenocarcinoma cases, occurring more frequently in women of Asian ancestry and never-smokers. TKIs targeting *EGFR* mutations include the first-generation drugs gefitinib and erlotinib, and afatinib and osimertinib, second and third generation, respectively [5]. Although TKIs improved progression-free survival, mechanisms of acquired resistance are common and lead to disease progression in most patients [5]. *ALK* rearrangements are found in ~7% of patients with lung adenocarcinoma and are mutually exclusive with *KRAS* and *EGFR* mutations [6]. The identification of the *ALK* rearrangement is predictive for therapeutic targeting by crizotinib. New generations of ALK TKIs ceritinib, alectinib, and brigatinib have been used [5]. *ROS1* rearrangements occur in 1–2% of cases and are most common in adenocarcinoma patients, younger patients (<40 years old), females, and never-smokers. Crizotinib is used for patients with ROS1-positive tumors [6]. Somatic mutations in *BRAF* occur in ~3–8% of adenocarcinoma cases; of these, ~50% are *BRAF* V600E, which is predictive for vemurafenib and dabrafenib-based therapies [5]. Genomic studies have revealed additional driver mutations in NSCLC, which can be explored for targeted therapy.

Conversely, patients with squamous cell lung carcinoma, which represents about 20–30% of NSCLC, have limited treatment options. Treatment with biomarker-driven therapies targeting FGFR, PI3K, MET, EGFR, among others, failed to demonstrate activity in the Lung Cancer Master Protocol (Lung-MAP SWOG S1400). However, an ongoing phase 2 open label clinical trial (RAGNAR) showed evidence of efficacy for erdafitinib, a selective pan-FGFR tyrosine kinase inhibitor, in heavily pretreated patients with different *FGFR*-positive solid tumors, including squamous and non-squamous cell lung cancer [14].

In the past decade, immunotherapy based on immune checkpoint inhibitors (ICIs) has shown significant survival benefits for patients with advanced NSCLC. Cancer cells develop immune evasion mechanisms playing a pivotal role in cancer progression. Monoclonal antibodies, such as pembrolizumab and nivolumab, are directed to block the PD-1 receptor in T lymphocytes, preventing immune response inhibition [1,15]. Patients with advanced NSCLC treated with ICIs have improved survival in comparison to standard chemotherapy in both first- and second-line therapies. The efficacy of nivolumab monotherapy in pretreated advanced non-squamous and squamous cell lung cancer showed a 17% objective rate response (ORR) and a median of 17.0 months of response duration among patients [16]. The combination of different ICIs with distinct and complementary mechanisms to improve anti-tumor immunity, such as nivolumab targeting PD1 and ipilimumab targeting CTLA4 in T lymphocytes, was tested in a phase 1, multi-cohort study showing high response rate and durable response with tolerable safety in NSCLC [17]. However, despite durable responses, not all patients benefit from ICI treatment [15], highlighting the importance of identifying biomarkers of immunotherapy response.

In this review, we describe the current state of knowledge regarding the molecular and cellular mechanisms involved in metastatic spreading of lung cancer cells to the brain. We discuss the influence of the brain microenvironment, including immune cells to support tumor cell growth. Moreover, a comprehensive discussion of genomic and transcriptomic alterations, including coding and non-coding RNAs, as genetic determinants of brain metastasis in NSCLC is presented. We also provide an overview of the current therapeutics, new treatment opportunities, and future directions for patients diagnosed with BM from NSCLC.

## 2. NSCLC Brain Metastasis

NSCLC frequently metastasizes to bone, brain, lung, and liver. BM accounts for most of the central nervous system (CNS) tumors, being observed in up to 40% of patients with different cancer types. Strikingly, BM is about 10 times more common than primary tumors affecting the CNS [18,19]. Patients with lung cancer have the highest rates of identified BMs [20]. Approximately 10–20% of NSCLC patients have BM at the time of diagnosis and approximately 40% will develop BM during the course of disease [21,22]. BMs often appear as multiple lesions, although one-third of patients present single lesions [23]. BM is highly prevalent in lung adenocarcinoma, markedly worsening patient outcomes, with a median survival of up to 15 months for treated patients [24].

The incidence of BM is probably underestimated since routine brain magnetic resonance image (MRI) screening in patients who do not present neurological symptoms is not recommended. Routine brain MRI would increase the detection of asymptomatic brain metastasis. However, its use as a populational guideline is controversial due to the high burden on the patients and the health care system [25,26]. In addition, a proportion of patients with negative screens may develop brain metastasis within one year [27]. Therefore, current guidelines support routine neuroimaging scans for more advanced clinical stages. Moreover, many studies frequently report the detection of BM at the time of initial diagnosis, but no information is provided on the subsequent sites of metastatic involvement during the disease course [23]. BM is often associated with severe neurologic and cognitive difficulties that are responsible for patient morbidity and significantly decreased quality of life. Headache, followed by neurologic dysfunction, seizures, stroke-like symptoms, and/or subtle cognitive dysfunction are the most common symptoms [19]. In fact, BM is often detected based on neurological symptoms without a diagnosis of a primary lung tumor. Studies have reported that most BM originating from lung cancer is located in the supratentorial area of the brain [28] and its distribution depends on the mutational status of *EGFR* [29].

Leptomeningeal metastases are a subset of BMs that grow in the lining of the brain or spine and/or cerebrospinal fluid (CSF) and occur with or without brain parenchyma metastases. Leptomeningeal metastasis is less common, occurring in 3–5% of patients with advanced NSCLC, and has been recently reviewed elsewhere [30]. Its incidence has increased in subgroups of patients who have received targeted molecular therapy due to the extended survival time. The prognosis of patients with leptomeningeal metastasis from NSCLC is poor; however, it has improved from a median historical survival (pre-approval of contemporary systemic treatment) of 1–3 months to 3–11 months with the use of new therapies [30,31].

The prognosis of patients with BM depends on different factors such as primary tumor site and other prognostic indicators, including driver mutations. In lung adenocarcinoma, BM occurs in ~20–40% of patients with *ALK* rearrangements and ~25% of patients with *EGFR*-mutated tumors [32,33,34]. The graded prognostic assessment (GPA) is a prognostic index that helps estimate patient survival in the presence of BM. In addition, patient age, Karnofsky performance status (KPS), extracranial metastases, and number of BMs are diagnostic-specific prognostic indices for patients with NSCLC. GPA scores range from 0-4, from worst to best prognosis, and define survival times ranging from 3.0–14.8 months for NSCLC patients [35]. An update of the GPA prognostic index including molecular markers, the Lung-molGPA, added *EGFR* and *ALK* mutation status for patients with lung adenocarcinoma. Median patient survival ranged from 3.0–46.8 months, although only 4% of the patients showed the highest scores (3.5–4.0) with a median survival of ~46 months [36]. Extensive efforts have focused on predicting outcomes for patients who develop BMs.

## 3. The Development of Brain Metastasis Is a Complex, Multistage Process

The major steps of metastatic spread to the brain are the dissociation of cells from the primary tumor, invasion of surrounding stroma and basement membrane, cancer cell intravasation, extravasation, and breaking down of the blood–brain barrier (BBB) followed by CNS invasion and colonization [18]. BM arises through seeding of circulating tumor cells (CTCs) into the brain microvasculature. Tumor cells interact with the brain endothelium, increasing the adhesion of tumor cells and promoting circulatory arrest. Once trapped, tumor cells start the process of crossing the BBB, which is a crucial step in tumor dissemination to the brain. The BBB harbors tight and adherens junctions between the brain endothelial cells, which regulate the flow of ions and nutrients, establishing a selective permeability barrier that protects the brain from blood-derived toxins and restricts the migration of leukocytes and monocytes [37,38,39]. BBB permeability is highly increased during BM in lung cancer [40], allowing CTCs to penetrate the brain and promote BM development. Several mechanisms associated with BBB remodeling that facilitate the migration of tumor cells through the BBB have been identified, including the secretion of various proteases to degrade tight junction components [41,42,43]. For example, cancer cells overexpress enzymes associated with mitogenesis and growth factors, including prostaglandin-endoperoxide synthase 2 (COX2) and heparin-binding EGF-like growth factor (HBEGF), allowing cell migration through the BBB [44,45]. 

Interestingly, tumor cells are also able to increase the expression of cathepsin S, a protease that is predominantly expressed by leukocytes, to cleave the junctional adhesion molecules that maintain BBB integrity and thus help tumor cells to break down the BBB [46]. In addition, extravasation of tumor cells, seeding, and formation of micrometastases are mediated by a combination of circulating proteins, including vascular endothelial growth factor (VEGF), matrix metalloproteinases (MMPs), among others, which are produced by tumor cells or cells in the tumor microenvironment (TME). Therefore, metastatic formation is mediated by a combination of circulating molecules, mainly proteins secreted by tumor cells and cells in the TME [23,47] (Figure 1).

To relocate to the CNS, disseminated circulating tumor cells (CTCs) must adapt to a microenvironment that is fundamentally different from the primary site. Immune cells, astrocytes, microglia, and neurons form a highly complex and dynamic TME, able to influence the survival of tumor cells and to modulate immune responses driven by metastatic brain cells [46,48]. The interaction between metastatic cells and the TME is critical for growth after cell seeding [23]. There is a complex reciprocal communication between metastatic tumor cells and their TME, which primes the successful outgrowth of cancer cells to form metastasis [45,49]. Astrocyte-derived exosomes mediate an intercellular transfer of *PTEN*-targeting microRNAs to metastatic tumor cells. As a consequence of PTEN loss, there is increased secretion of C-C motif chemokine ligand 2 (CCL2), which in turn induces recruitment of IBA1+ myeloid cells, enhances brain metastatic tumor cell proliferation, and reduces apoptosis [49]. The loss of BBB integrity is also a result of neuroinflammation and direct rupture of the barrier by tumor cells. Metastatic cells interact with neuroinflammatory cells and other components of the brain parenchyma, leading to tumor colonization. Secondary tissue colonization is a main bottleneck in metastatic development. Nonetheless, the pre-metastatic stage of the metastatic cascade remains poorly characterized. At the moment, studies using brain metastasis initiating cells (BMIC) show that the pre-metastatic stage of brain tissue colonization involves deregulated genes, many of which are active in invasive but not in proliferative mechanisms [50]. However, the process of metastatic brain colonization and changes in the microenvironment of metastatic tumors are not fully understood [23,51].

BMs, even in initial stages, are surrounded by a significant neuroinflammatory response mediated by activated astrocytes and microglia. Given the presence of the resident microglia and the lymphatic system, the brain is no longer considered a place with immunological privileges. The established metastases induce brain damage leading to the infiltration of immune cells, including CD8+ cytotoxic T lymphocytes. The expression of PD-1 and PD-L1 proteins in resected BMs indicates an immunosuppressive TME [23]. The extravasation of tumor cells, seeding, and formation of micrometastases is mediated by a combination of circulating proteins produced by tumor cells or cells in the TME. After extravasation, individual cancer cells are immediately surrounded by reactive astrocytes that act as an efficient first line of protection in the CNS by reducing the number of cells that initiate potential metastases. This natural defense contributes, in part, to the high inefficiency of colonization of the brain by cancer cells. Some cancer cells manage to survive and remain located in the perivascular niche next to the neural stem cells, where cancer cells have greater access to nutrients and oxygen, contact with the basal lamina of capillaries, and preferential access to angiocrine growth factors produced by endothelial cells [51].

The proliferation of cells that initiate metastases establishes a variable number of micrometastases. Some micrometastases can physically interact with reactive astrocytes. These interactions increase the growth of cancer cells and resistance to chemotherapy-induced apoptosis [51]. Astrocytes have also been shown to be critical modulators of immune responses in BM. They interact with inflammatory cells resident in the brain and are recruited along with the microglia, leading to the establishment of an immunosuppressive microenvironment [49,52]. Thus, astrocytes are emerging as one of the main regulators of colonization and metastatic growth in the brain [46,53].

Several studies have shown that at every step during the metastatic cascade, cancer cells must engage different metabolic strategies, distinct from the primary tumor, to successfully metastasize [54,55,56]. While normal brain cells depend on glucose for energy production, metastatic cancer cells in the brain possess metabolic flexibility and depend not only on glucose for energy, but also on glutamine and acetate [57]. These metabolic adaptations are the result of interactions between cancer cells and brain cells including astrocytes and neurons, which promote rapid metastatic growth in the brain [57,58,59].

## 4. Molecular Determinants of Brain Metastasis in Lung Cancer and Their Implications for Treatment

Increasing evidence suggests that metastasis results from the aberrant activation of “invasive growth”, a morphogenetic program that occurs during embryonic development and postnatal organ regeneration, driven by the *MET* proto-oncogene [60,61]. *MET* has been shown to play a central role in BM from lung cancer [62,63]. Additionally, Recepteur d’Origine Nantais (*RON*), also known as macrophage stimulating receptor 1 (*MSTR1*), a member of the MET family of receptor tyrosine kinases, harbors somatic mutations that are predicted to cause deleterious effects in BM from lung carcinoma [64]. Hyperactivation of the WNT/TCF signaling pathway has also been associated with BM formation in lung adenocarcinoma, mainly through the altered expression of the transcription factors HOXB9 and Lymphoid Enhancer Binding Factor 1 (LEF1), which stimulate tumor cell invasion and proliferation [65]. 

Studies have investigated metastatic genomic profiles in lung cancer. Genomic alterations in cancer-related genes in primary and matched metastatic tumors from 15 NSCLC patients, including 8 lung adenocarcinoma tissues [66], showed a concordance rate of 94% of recurrent alterations between primary tumor and matched metastasis, with *TP53* mutations being the most frequently observed [66]. Genomic characterization of stage IV lung squamous cell carcinoma of 79 patients reported hot-spot mutations in 8 main oncogenes (*EGFR*, *KRAS*, *BRAF*, *PIK3CA*, *NRAS*, *HER2*, *MEK1*, and *AKT1*) as well as exonic and intronic mutations of 279 cancer-related genes [67]. The data were also analyzed according to two molecular subtypes: cases with fibroblast growth factor receptor 1 (*FGFR1*) amplification or mutation or loss of upstream phosphoinositide 3-kinase (PI3K) pathway genes, i.e., *PTEN* and *PIK3CA*. BMs were present in 11% (9/79) of cases, all from patients with PI3K altered tumors (27%; 9/33 patients). Six of the nine BM cases were further investigated by whole-exome sequencing (WES), RNA sequencing, and immunohistochemistry, and compared with a subset of four corresponding primary lung tumors. Results showed *PTEN* loss in 4/6 BMs and in all four primary tumors.

In addition, genetic alterations driving BM formation/progression were previously reported. Whole-exome sequencing of 73 BM cases from lung adenocarcinoma (BM-LUAD) identified *MYC*, *YAP1*, *MMP13* amplifications and *CDKN2A/B* deletions as pathogenic genomic changes [68]. Additionally, it was demonstrated that overexpression of these candidate driver genes (*MYC*, *YAP1*, or *MMP13*) promoted BM in patient-derived xenograft (PDX) mouse models [68]. In another study, by comparing focal somatic copy number alterations (SCNAs) in matched NSCLC-BM pairs, putative BM-driving genetic alterations were identified affecting multiple cancer genes, including several potentially targetable changes in genes such as *CDK12*, *DDR2*, *ERBB2*, and *NTRK1* [69]; these results were validated in an independent cohort of 84 BM samples and characterized SCNAs and mutations in *EP300*, *CTCF*, and *STAG2* genes, which play roles in epigenome editing and 3D genome organization [69]. Whole exome sequencing analysis of 12 paired primary NSCLC and matched BM have also identified BM-associated mutations in known cancer genes including *AHNAK2*, *ANKRD36C*, *BAGE2*, *KMT2C*, and *PDE4DIP* [70].

BMs may harbor high genetic heterogeneity and clonal differences between their corresponding primary tumors, suggesting that additional molecular changes may be acquired during metastatic progression [67]. Several studies have been performed in an attempt to address the question of clonality and molecular heterogeneity between primary tumors and brain metastasis from same patients. Some studies have collected and profiled metastatic lesions in an asynchronous mode with the primary tumor, allowing detection of evolutionary changes over time. In a report by Lee et al. [71], multi-omics sequencing of seven paired tumors and BM (collected at different time points) from patients with NSCLC, showed that 67% of mutations were common between metastatic and primary samples. In addition, these lesions had a similar tumor mutational burden (TMB). Further validation using publicly available data from a whole exome sequencing study of 35 BM and primary samples [72] showed 69% of shared mutations and similar TMB frequency. Based on these findings, the authors suggested that metastatic events occur late during the evolutionary tumor development and progression cycles, likely upon the establishment of the majority of somatic mutations in the primary tumor [71]. Although the results of this study are based on a small sample set of 7 patients, the authors also suggested that a monoclonal mode of metastatic seeding may be predominant in most NSCLC cases.

Interestingly, Brastianos et al. [72] identified that, although there are genetic similarities between BM lesions arising in different brain sites as well as temporally separated, there is high genetic heterogeneity between BM and lymph node metastasis or extracranial distant metastasis. In addition, they reported actionable changes in BM, correlated with drug sensitivity to PI3K/AKT/mTOR, CDK, and HER2/EGFR inhibitors [72]. Other studies found similar results [73] and reported molecular changes likely selected during metastatic progression, such as deletions of *CDKN2A/B* which were common to metastatic and primary samples [68]. A recent whole exome sequencing study of 84 tissue samples from 26 patients compared genomic profiles of primary lung adenocarcinoma, liver and BM lesions; this study showed common driver mutations in *TP53* and *EGFR* in primary and metastatic samples. Additionally, a comparable TMB was present in all samples; however, the liver metastases had higher TMB and were more similar to the primary tumors than the BM lesions [74]. These authors also performed phylogenetic analyses and found that liver metastasis was genetically divergent from the paired primary tumors at a later stage of metastatic development compared to BM sites, suggesting that liver metastasis may arise preferably through a linear mode and BM may be established following a parallel mode of progression. It is important to highlight some differences among published studies, which may be due to different patient cohorts, distinct methodologies of sample collection with metastatic samples being collected either synchronously or asynchronously with the primary tumors, and different platforms of analyses. Although the current knowledge on the genetic divergence and phylogenetic evolutionary relationships among BM lesions and primary tumors, this is still an area that deserves further and detailed investigation.

Genetic studies have been performed to characterize BM. Examples include a targeted panel of 160 cancer-associated genes assessed in 39 lung adenocarcinoma patients with synchronous BM (*n* = 10, BM tissue only), metachronous BM (7/12 paired primary tumor biopsies and BM tissues) or extracranial metastases [75]. Results from this study showed aberrations affecting genes in the PI3K/AKT signaling pathway in primary and BM tissues. Comparing BMs versus extracranial metastases, *BCL6* and *NOTCH2* variants were the only variants identified in at least four patients with BMs while appearing in only one or none of the non-metastatic cases. Unique variants were also detected in 20 genes (*TP53*, *SMAD4*, *SF3B1*, *NOTCH2*, *mTOR*, *MSH6*, *KRAS*, *KMT2D*, *KDM6A*, *IKZF1*, *GNAS*, *FANCD2*, *ERCC5*, *EP300*, *CREBBP*, *CDK12*, *BRCA2*, *BCL6*, *ATM*, and *ABL1*) in >33% of patients with BMs. Although there is a need for additional validation, this panel of genes was suggested to discriminate against the risk of developing BM [75]. More recently, tumor samples from 91 NSCLC patients (32 of which developed BM) were analyzed by Illumina RNA-sequencing. This study identified 22 genes (including *CELF1*, *NEURL2*, *CEBPB*, *AANAT*, *TMEM121*, *TWIST2*, *TNN*, *ST6GAL2*, *SLC38A4*, *FFAR4*, *LRIG3*, *CYB561*, *DPP9-AS1*, *C5orf60*, *ZNF843*, *ANKRD62*, *ZNF439*, and *PSG2*) that specifically correlated with BM and not with metastasis to other sites [76].

NGS targeted sequencing of 416 cancer-associated genes in primary lung tumors (*n* = 61 patients: 50 adenocarcinoma, 3 squamous cell carcinoma, and 8 mixed histology) and paired BMs showed that mutations in *EGFR*, *KRAS*, *TP53*, and *ALK* were concordant between primary tumors and BMs in >80% of cases. Of these patients, 25 patients (41%) had synchronous BMs, which showed a larger number of cancer-associated mutations when compared to primary tumors; this was not observed for patients with metachronous BM [77]. These data suggest that synchronous BMs are likely to undergo genomic evolution with the activation of additional oncogenic mechanisms [77]. Another study with seven lung adenocarcinoma patients with BM identified that the protein expression levels of TYMS, CDK1, HJURP, CEP55, and KIF11 were highly predictive of BM in these patients [78].

Studies such as those outlined above show the existence of selective molecular mechanisms driving clonal evolution of BMs. Clones of metastatic cells growing in the brain evolve alternative routes, compared to other metastatic sites, mainly due to the hostile nature of the brain microenvironment for cancer cells. These data also have important implications to delineate tailored therapeutic approaches and to develop robust, clinically applicable biomarkers to identify patients at high risk for BM [23,51].

Effective treatments for patients with BM need to consider the molecular changes specific to metastatic cells, as well as the BM microenvironment [23]. Indeed, targeted therapies based on primary tumor driver mutations might fail to treat patients with BMs, due to the molecular divergence between BMs and primary tumors [34]. Current international recommendations include the identification of molecular alterations specific to BMs in tissues and in liquid biopsies, which may be clinically applicable for early detection of BMs, and detection of BM-specific molecular changes to evaluate therapeutic response [34].

Considering the reported evolutionary changes and genetic differences between primary tumors and BM, as well as the molecular features that result from selective pressures of systemic treatments, there is a need for additional molecular profiling studies leading to biomarker development in BM. The relevance of novel biomarker discovery in BM and also extracranial metastasis includes the identification of molecular determinants/drivers of metastatic progression and the application of this knowledge for more effective treatment approaches to improve patient survival [79].

Recent studies applying single cell transcriptomic sequencing were able to unlock novel key molecular features of individual metastatic cells in lung cancer. Among these, Ruan et al. [80] characterized transcriptome changes in circulating tumor cells in cerebrospinal fluid of patients with lung adenocarcinoma leptomeningeal metastasis. They sequenced 792 transcriptomes of 5 patients and 3 controls and found a metastatic signature of genes with roles in metabolism as well as molecules related to cell adhesion. Furthermore, this study reported that there is higher heterogeneity among patients when compared to single cells isolated from the same patient [80]. In addition, Zhang et al. [81] identified differentially expressed genes between primary lung adenocarcinoma and BM isolated from two patient-derived xenografts (PDX). The authors suggested that *CKAP4* (Cytoskeleton Associated Protein 4), *SERPINA1* (Serpin Family A Member 1), *SDC2* (Syndecan 2), and *GNG11* (G Protein Subunit Gamma 11) are potential biomarkers to aid in prognosis assessment and therapy of patients with lung cancer BM [81].

Other technologies were applied for digital spatial RNA sequencing profiling of NSCLC and BM, and allowed a comprehensive assessment of biomarkers associated with primary and metastatic lesions, including analysis of the primary tumor immune and the brain microenvironments. Zhang et al. [82] analyzed a cohort of 44 patients with metastatic NSCLC using the NanoString GeoMx DSP platform. Among their findings, we highlight the highly immunosuppressive microenvironment associated with BM lesions compared to primary tumors, with reduced abundance of B and T-cells and higher infiltration of neutrophils. Their study shed light on the role of molecular changes in the tumor and BM microenvironments for establishment of the metastatic niche [82]. Studies such as these are relevant to determine clinically applicable biomarkers for patient treatment.

## 5. Non-Coding RNAs Play Important Roles in Brain Metastasis from Lung Cancer

Non-coding RNAs (ncRNAs) are regulatory molecules that modulate several biological functions, including gene expression, cell signaling, and genomic rearrangement [83,84]. NcRNAs are classified based on their lengths: small ncRNAs (sncRNAs; <200 nucleotides) such as microRNAs (miRNAs) and PIWI-interacting RNAs (piRNAs), and long noncoding RNAs (lncRNAs; >200 nucleotides) [85,86]. NcRNAs play roles in cancer progression, with specific expression patterns during metastasis development [87]. Alterations in ncRNA expression levels, mainly in miRNAs, could be associated with the development of BM in lung cancer since miRNAs have important regulatory roles in different steps of the metastatic cascade, including migration, invasion, adhesion, colonization, and epithelial-mesenchymal transition (EMT) [88,89,90]. MiRNAs can also contribute to disrupting the BBB [91] by creating a more hospitable environment for metastatic-initiating cells [92], establishing a pro-metastatic microenvironment [89,93] and modulating cancer stem cell (CSCs) properties that could contribute to the establishment of BM [94].

The mechanistic roles of oncogenic miRNAs in the development of BM from NSCLC have been explored. MiR-328 deregulation has been shown to promote BM in patients with NSCLC, partially by modulation of protein kinase C alpha (PRKCA), leading to high PRKCA levels and increased cancer cell migration [95]. MiR-378 has also been shown to be overexpressed and associated with an increased risk of BM and “brain-seeking” metastatic potential, due to its role in promoting cell migration, invasion, and tumor angiogenesis through modulation of *MMP-2*, *MMP-9*, *VEGF*, and suppressor-of-fused (*SUFU*) genes [96]. *SUFU* is involved in glioma cell growth and angiogenesis [97], and metastasis in lung adenocarcinoma [98]. Another report showed that miRNA-197 and miRNA-184 are overexpressed in *EGFR*-mutant BM when compared with *EGFR*-mutant primary tumors without BM [99]. However, this study did not include *EGFR* wild-type tumors from patients with and without BM for comparison with *EGFR*-mutant cases. Hence, more research is needed to understand how these miRNAs affect *EGFR* regardless of its status.

Tumor-suppressive miRNAs have also been implicated in metastatic progression. In animal models, miRNA-146a was suppressed in BM compared to primary tumors [100]. Overexpression of miR-146a suppressed the metastatic potential, including migratory and invasive activities, through upregulation of β-catenin and downregulation of heterogeneous nuclear ribonucleoprotein C1/C2 (hnRNPC) [100]. In line with these findings, miRNA-95-3p is downregulated in BM of lung cancers compared to primary tumors [101]. Overexpression of miRNA-95-3p suppresses cyclin D1 expression through direct binding to the 3′ UTR of cyclin D1 mRNA and suppresses invasiveness, proliferation, and clonogenicity in in vitro assay [101]. Similarly, downregulation of miRNA-145 [102], and miRNA-375 [103], has been associated with BM formation in NSCLC; evidence suggests that while miR-145 overexpression reduces cell proliferation, there is no effect on the migration and invasion ability of cell lines. This indicates that miR-145 downregulation likely enhances cell proliferation after having reached the metastatic brain site, aiding in colonization, rather than in the early stages of metastasis [102].

Another miRNA found to be underexpressed in BM, compared to matched primary lung cancer tissues, is miRNA-768-3p [104]. Subramani et al. suggested that the brain microenvironment negatively regulates miRNA-768-3p, which enhances *KRAS* expression contributing to metastasis [104]. In another study with a cohort of 357 stage I NSCLC patients, 10 miRNAs correlated with BM (hsa-miR-450b-3p, hsa-miR-29c, hsa-miR-145, hsa-miR-148a, hsa-miR-1, hsa-miR-30d, hsa-miR-187, hsa-miR-218, hsa-miR-708, and hsa-miR-375) [105]. Taken together, these findings suggest that the loss of miRNAs with a tumor suppressive role could activate oncogenic pathways that are hallmarks of cancer, contributing to tumor development and progression to metastasis.

LncRNAs also play fundamental roles in lung tumorigenesis and metastasis [106,107]. They have been shown to modulate chromatin functions, control membraneless nuclear bodies’ assembly and function, alter cytoplasmic mRNA stability and translation, and interfere with signaling pathways, depending on their localization and specific interactions with DNA, RNA, and proteins [108]. Examples of lncRNAs associated with lung cancer BM include the metastasis-associated lung adenocarcinoma transcript 1 (*MALAT1* or nuclear enriched abundant transcript 2, *NEAT2*), which is overexpressed in a variety of tumors, including metastatic NSCLC [109]. *MALAT1* acts through promoting migration of cancer cells to the brain in an EMT-driven mechanism [109]. Furthermore, *MALAT1* promoted migration and invasion by targeting miR-206 and activating Akt/mTOR signaling in NSCLC tissues and cell lines [110]. Another lncRNA known in tumor development and progression, HOX transcript antisense RNA (*HOTAIR*), was associated with BM from NSCLC [111]. In vitro studies showed that *HOTAIR* enhances cell migration and anchorage-independent cell growth [111]. Nonetheless, the exact role and target of *HOTAIR* remains unknown.

Recently, it was also reported that the histocompatibility leukocyte antigen complex P5 (*HCP5*) is a potential driver for BM in lung cancer [112]. Computational bioinformatic analyses suggested that the ferroptosis-related competing endogenous RNA (ceRNA) *HCP5*/miR-17-5p/*HOXA7* axis may contribute to the development of BM in lung adenocarcinoma [112]. In addition, overexpression of AC122108.1 lncRNA promotes BM in lung adenocarcinoma through the Wnt/β-catenin pathway by directly binding to the aldolase A (ALDOA) protein; this mechanism enhances the proliferation, apoptosis, invasiveness, migration, and metastasis of lung adenocarcinoma cells [113]. In patients with limited-stage SCLC, a recent study using peripheral blood mononuclear cells (PBMCs) identified the low-level expression of lncRNA XR_429159.1 as a high-risk factor for BM [114]. However, the underlying mechanisms need to be further explored. Other lncRNAs involved in promoting metastasis in lung cancer include chromatin-associated RNA 10 (*CAR10*) [115] and brain cytoplasmic RNA 1 (*BCYRN1*) [116].

Recent studies have shown that lncRNAs are closely related to the increased permeability of the BBB in BM development and brain tumors [117,118,119,120]. In NSCLC, both exosomal-derived LINC01356 and lnc-MMP2-2 were found to increase BBB permeability and promote BM development. While the exosomal lncRNA LINC01356 remodels BBB by targeting cell junction proteins such as Occludin, Claudin, and N-cadherin [121], TGF-β1-mediated exosomal lnc-MMP2-2 may destroy the tight junctions of the BBB, thereby facilitating the passage of cancer cells [122].

Moreover, lncRNAs may interact with immune cells in the brain and contribute to a permissive environment for tumor growth [123]. For example, glioma cell-derived exosomes are able to transport lncRNA-ATB to astrocytes, promoting their activation, which in turn facilitates invasion and migration of glioma cells [124]. In breast cancer cells, loss of lncRNA X-inactive-specific transcript (XIST) triggers the polarization of microglia, resulting in increased expression of cytokines and suppression of T-cell proliferation [125]. Immune suppression is one of the mechanisms by which microglia promotes tumor progression in the brain [125]. Another study showed that JAK2-binding long noncoding RNA can promote breast cancer brain metastasis through a STAT3-dependent mechanism, which mediated recruitment of macrophages into the brain [126]. In addition, a recent report demonstrated that lncRNA (BMOR) is important for developing breast-to-brain metastasis by allowing tumor cells to evade immune-mediated killing in the brain microenvironment [127]. Altogether, these studies evidence the importance of lncRNAs for mediating communication between cancer cells and the brain microenvironment. Given the importance of these lncRNAs, it would be interesting to explore whether it is also involved in lung-to-brain metastasis. Figure 2 illustrates known miRNAs and lncRNAs involved in several important steps of BM development in lung cancer.

Strategies that identify and target miRNAs and lncRNAs may be attractive as early diagnostic and therapeutic options. In recent years, the deregulation of ncRNAs in lung cancer has prompted preclinical studies examining the therapeutic potential of restoring and/or inhibiting such molecules [128]. The tissue-specific expression as well as high stability within body fluids makes them excellent candidates as biomarkers for diagnosis, prevention, and treatment of BM [128]. It has recently been demonstrated that miRNAs can be used to distinguish normal cells from cancerous ones and primary brain tumors from secondary brain tumors, as well as correctly categorize metastatic brain tumor tissues based on their expression profiles [93]. These data indicate that miRNAs are promising candidates for clinical applications in BM. On the other hand, the roles and molecular mechanisms of many lncRNAs still remain elusive. Though promising, several challenges remain to be addressed to implement ncRNAs in clinical practice [93,129,130], such as the development of efficient delivery systems capable of crossing the BBB, with minimal toxicities, and the successful unloading of ncRNA therapeutics.

## 6. Advances in the Molecular Diagnostics of Brain Metastasis: Liquid Biopsies

Liquid biopsy is a new diagnostic concept based on the analysis of circulating tumor cells (CTCs), and/or molecules originated or secreted by tumor cells. Molecules derived from body fluids that are useful for liquid biopsy tests include circulating tumor DNA (ctDNA) and RNA (ctRNA), proteins, and microvesicles (e.g., exosomes) [131]. The ctDNA and ctRNA (coding and non-coding) are passively released from apoptotic or necrotic tumor cells, or are actively secreted by cancer cells [131].

Liquid biopsies are a minimally invasive alternative to tissue biopsies, particularly for tissues that are difficult to obtain, such as the brain. Furthermore, liquid biopsies can be serially repeated since they are minimally invasive and have low cost [132]. Liquid biopsies have an enormous potential to monitor treatment response, quantify minimal residual disease, and assess the emergence of clones resistant to therapy (Figure 3). Several types of body fluids are useful in the development of liquid biopsy diagnostic tests in cancer: blood, pleural effusion, and CSF [131]. Cancer-specific changes can be measured in liquid biopsies, including genomic, transcriptomic, proteomic, and CTC quantification. Translation of liquid biopsy tools into clinical practice is transforming diagnosis in oncology, as demonstrated by a large number of liquid biopsy diagnostic tests entering into the clinical setting [133]. Indeed, the first approved commercial liquid biopsy test detects *EGFR* mutations in ctDNA, and it is useful to select metastatic NSCLC patients for EGFR-TKIs [134,135].

CTCs are isolated or clustered tumor cells released by the primary tumor or metastasis that leaks into the bloodstream and migrates towards the metastatic site. The frequency of CTCs is found on the order of 1–10 CTCs/mL of whole blood in patients with metastatic disease [136]. Different methods for enrichment, isolation, and identification of CTCs were developed according to their physical and biological characteristics. In lung cancer, isolation by size of epithelial tumor cells (ISET) was the earliest size-based method used for CTC detection, showing high sensitivity and reproducibility [137]. High levels of CTCs have been associated with worse outcomes in lung cancer. In 2017, Lindsay et al. evaluated the total number of CTCs as a prognostic marker in 125 treatment-naive patients with advanced NSCLC. Vimentin-positive CTCs were assessed according to treatment and the presence of *EGFR*, *ALK,* and *KRAS* mutations; a number higher than 5 CTCs was associated with reduced survival and an increase in vimentin-positive CTCs was associated with EGFR-mutated tumors, suggesting the presence of epithelial–mesenchymal transition characteristics [138]. Vimentin is a filamentous protein expressed in mesenchymal cells, and it is known to maintain cellular integrity and provide resistance against stress [139]. It has been often recorded in cancers, including NSCLC and BM, as a marker of tumor cell invasion via its expression during the aberrant activation of epithelial–mesenchymal transition (EMT) [138]. During EMT, vimentin modulates genes for EMT inducers, as well as some key epigenetic factors. It suppresses cellular differentiation and upregulates their pluripotent potential by inducing genes associated with self-renewability, which increases the stemness of cancer stem cells, facilitating tumor spread and making tumor cells more resistant to treatments [140]. Vimentin overexpression has been associated with increased cancer cell growth, invasion, and migration, suggesting its potential application in cancer diagnosis and treatment [139]. Another study showed that the presence of CTCs was associated with low response rates, as well as shorter progression-free and overall survival, in patients with advanced NSCLC treated with both targeted- and chemotherapy [141]. CTCs derived from brain metastases were shown to have mutations in adaptive, cytoprotective genes with roles in the Keap1-Nrf2-ARE pathway, helping metastatic-initiating CTCs to survive in the blood circulation [142]. Therefore, CTCs may be an ideal source for determining the molecular portrait of metastasis where tumor biopsies are not clinically feasible. Importantly, CTCs can be expanded in vitro and in vivo, the latter by establishing CTC-derived xenografts as a means to characterize CTCs capable of initiating metastasis. Such applications have the potential to demonstrate the molecular mechanisms of metastasis initiation driven by CTCs and are promising in the discovery of novel molecular diagnostic and therapeutic strategies [143]. 

CTCs detected by liquid biopsies will help understand the molecular aspects of metastatic progression. Indeed, CTCs have been indicated as clinically applicable for many years, and a plethora of studies have demonstrated the correlation between CTC counts and metastatic disease in different cancer types [144]. In NSCLC patients with BM, CSF has been shown to be useful for CTC detection [145,146]. Recent functional studies have established in vitro and in vivo models from CTC-derived metastatic cells and are valuable to reveal molecular alterations specific to aggressive, metastatic-enabling clones [143]. Further development of methods for detection of metastasis-initiating CTCs will help elucidate the processes by which metastasis is established into the brain and extracranial sites. Darlix et al. [147] reported a prospective study for detection of suspected leptomeningeal metastasis in 40 patients with breast cancer. The authors tested the CellSearch^®^ system, a clinically validated and FDA-approved test for CTC detection [147] and were able to identify CTCs in the cerebrospinal fluid of all cytology-positive samples. Interestingly, they detected CTCs in five cytology-negative samples, demonstrating improved sensitivity of CTC detection using the CellSearch^®^ system. Furthermore, they were able to detect HER-2 positive CTCs in the CSF of HER-2 negative tumors. This same system has been previously used to evaluate detection of both CTCs and exosomes in pancreatic cancer patients and was shown useful to accelerate diagnosis and treatment of surgically resectable cases [148]. Such studies highlight the importance of liquid biopsies as a potential tool to study molecular changes specific to more aggressive circulating tumor cells, as well as to refine molecular diagnostics and aid in treatment decisions that will impact patient survival [147]. Furthermore, the development of liquid biopsy-CTC-based biomarkers can be useful as a complementary tool to aid diagnostic imaging, augmenting early detection and, consequently, treatment intervention of occult BM [149].

Among cancer biomarkers, the proteome is a major source of circulating molecules that can inform clinically useful decisions [150]. A few examples include the circulating protein biomarkers CEA, PSA, β-hGC, CA 15-3, and CA 19-9 [151]. Carcinoembryonic antigen (CEA) is the most studied biomarker in lung cancer, investigated as a prognostic biomarker for BM development. High CEA serum levels were associated with BM in NSCLC [152]. CEA was shown as a prognostic biomarker for BM, as well as cytokeratin 19 fragments (CYFRA 21-1), cancer antigen 125 (CA125), cancer antigen 19-9 (CA19-9), and squamous cancer cell antigen in NSCLC [153]. Cancer antigen 125 has been used as a clinical tumor marker for prognosis and therapy monitoring in ovarian and breast cancer patients [154]. However, other studies reported CA-125 as a marker for worse prognosis in lung cancer [155] and a prognostic biomarker in BM [153]. In addition, high serum levels of lactate dehydrogenase (LDH), CEA, CYFRA 21-1, and CA125 were independently associated with BM in a large cohort of geriatric patients with lung adenocarcinoma [156]. Distinct from the genome, the proteome composition can change in response to variations in intracellular and extracellular conditions. Considering that gene expression implicates alternative splicing and post-translational modifications, the number of expressed proteins vastly outnumbers the number of genes. Therefore, proteome analysis can uncover molecular pathways, protein–protein interaction networks, and events underlying cellular phenotypes associated with the disease. The evolving field of oncoproteomics will likely derive novel biomarkers ready to use in liquid biopsies for clinical practice applications [151].

## 7. Treatment of BM from Lung Cancer

Primary management of BM predominantly has consisted of local treatments including surgery, stereotactic radiation or large field radiation therapy based on the knowledge of the heterogeneous penetration of systemic therapies into the brain. More recently, advances in systemic treatment were developed, particularly with the introduction of molecularly targeted therapeutics and immunotherapies. Nevertheless, direct evidence of systemic therapy in BM is limited since the presence of BM is generally an exclusion criterion in randomized trials, or patients with BM are underrepresented in these trials [157,158,159,160]. A comprehensive review of treatment for BM in patients with NSCLC is available by Tsui et al., 2022 [160].

Treatment for brain metastases comprises two broad categories, of symptomatic management and tumor-directed therapies [23]. Corticosteroids, such as dexamethasone, represent the main treatment for symptomatic patients, frequently prescribed in response to signs of increased intracranial pressure due to peritumoral edema [23]. Anticonvulsants may be prescribed to prevent seizures, and systemic steroids alone may improve neurological function and prolong survival by approximately two months [161]. Most therapies may include a combination of surgery (aiming at diagnosis and brain decompression), and/or adjuvant radiotherapy or systemic therapies. Other modalities include Stereotactic Radiosurgery (SRS) and whole-brain radiotherapy (WBRT) (Figure 4). The approach adopted for a given patient will depend on the performance status, and the distribution of intracranial and extracranial disease [162,163]. Novel methods for minimally invasive neurosurgery were demonstrated to have advantages such as inexpensive instruments, straightforward use and operation, and accurate positioning. Modern technologies for minimally invasive surgery are suitable for clinical practice in medical institutions [164].

Corticosteroids have long been used to treat peritumoral edema. Their effects on improving symptoms are beyond the reduction of inflammation but include an upregulation of tight junction proteins (such as ZO-1 and occludin) [165], which are important in the maintenance of the blood–brain barrier’s structure and function. Endothelial cells within and around the tumor are damaged by the presence of the tumor and the corresponding inflammation, leading to increased permeability of vessels and extravasation of fluid [166]. Despite the advantages of corticosteroid use, their symptom relief is temporary and dependent on the control of the local disease. In addition, the side effects of prolonged use are well known, such as diabetes, weight gain, cushingoid features, hypertension, myopathy, and osteoporosis—with increased risk of fractures [167]. Therefore, the recommended dose is the minimum needed to control symptoms, varying from 4 mg/day to 16 mg/day depending on the patient’s symptoms [168].

The use of anticonvulsants for seizure prophylaxis is controversial. Despite being largely used in clinical practice, there is a paucity of high-level studies supporting their use. Current guidelines do not support the use of anticonvulsants for patients with newly diagnosed brain tumors without a history of seizures. For patients undergoing surgical removal of tumoral lesions (which could be at potential risk of developing epileptogenic foci), there is insufficient evidence to support the use of anticonvulsants within the perioperative period [169]. In the case of seizures, the most commonly used agents are phenytoin, levetiracetam, valproic acid, and carbamazepine.

Surgical resection of BM may provide important benefits: symptom relief, de-obstruction of CSF pathways, dismissal of corticosteroid use, and samples for histopathologic and molecular analysis [19]. Indications for surgical resection include single or few intracranial lesions (up to three), large lesions, and accessible sites (i.e., sites where surgical corridors will not determine new deficits). En bloc resection seems to decrease the risk of future leptomeningeal dissemination and is preferable to piecemeal resection, even though larger tumors may require some intraoperative internal decompression in order to prevent additional neurological deficits [170,171]. Moreover, gross-total resection may improve overall survival [172]. Neuronavigation and intraoperative ultrasound are helpful for precise tumor localization. As with other brain tumor surgeries, brain mapping, awake craniotomy, and intraoperative monitoring are good options for safer resection.

Overall median survival rates after surgical resection of BM range from 9.8 to 24 months [173,174,175]. Within one year, survival rates are about 40–45% [173,176]. Factors determining better outcomes and prolonged survival are lower number of BM, lower age, better preoperative clinical performance, absence of extracranial metastasis, and association with other treatment modalities (i.e., radiotherapy and immunotherapy) [173,174,175,176,177,178]. For single BM, survival can be longer than two years in cases of complete resection and smaller tumor size [179,180].

Radiation therapy plays an important role in the management of BM. Post-surgical radiotherapy may reduce the risk of local recurrence, even though it is not clear whether it changes overall survival (probably because of initial poor clinical conditions) [181]. WBRT is commonly used for multiple lesions inaccessible for surgical resection. However, it carries a significant risk of neurocognitive decline and diffuse leukopathy. Therefore, stereotactic radiotherapy or stereotactic radiosurgery are preferable, as they target the lesions without causing diffuse brain damage [182,183].

The molecular characterization of lung adenocarcinoma was pivotal to patient management. However, this knowledge for SCLC and the squamous subtype of NSCLC had a lower impact in patient treatment, mainly due to the low incidence or absence of targetable mutations in these tumor subtypes [159]. Current guidelines for BM treatment recommend targeted therapy only for those patients with oncogenic driver mutations [157]. Small molecule treatments have proven beneficial for palliative relief. For example, patients with NSCLC BM and positive for the *EGFR* mutation have shown meaningful CNS efficacy after treatment with third generation of EGFR TKIs such as icotinib [184] and osimertinib [185,186,187]. Similarly, intracranial response was observed in patients with BM treated with the third generation of TKIs targeting ALK rearrangements, alectinib [188], brigatinib [189,190], and ceritinib [191]. Other therapies have been explored for NSCLC, with BM showing promising results, such as capmatinib [192] and tepotinib [193], both targeting MET alterations; selpercatinib targeting RET fusions [194,195]; entrectinib [196], and larotrectinib targeting tropomyosin receptor kinase (TRK) fusion-positive tumors [197,198]; lorlatinib targeting *ROS1* [199]; and dabrafenib plus trametinib targeting *BRAF* V600E [200], among others [160,201].

The reduced penetration of chemotherapy agents through the BBB has limited its use as a primary treatment for BM in NSCLC [144]. Pemetrexed-cisplatin was shown to be effective for treatment of BM in patients with NSCLC with an objective response rate of 41.9% [182]. The FDA-approved drug Entrectinib was developed to target NTRK fusion as well as ROS and ALK tyrosine kinases and is capable of crossing the BBB. This drug has shown positive intracranial response rates in ROS-positive NSCLC [202,203], and NTRK fusion-positive solid tumors [196]. In addition, Lorlatinib, a potent brain-penetrant, third-generation tyrosine kinase inhibitor, has shown clinical activity in patients with advanced ROS1-positive NSCLC with BM [199]. Moreover, chemotherapy combined with immunotherapy has been shown to enhance the efficacy of immunotherapy, opening new windows for new first-line therapeutic strategies to benefit patients with advanced NSCLC [204,205]. 

Immunotherapy using ICIs is used in the management of NSCLC, particularly for patients without molecularly targetable disease. The benefit of ICIs in oncogene-targetable NSCLC is limited. Cumulative evidence suggests an interplay with tumor cell oncogenic signaling and tumor immunogenicity, leading to non-T cell-inflamed environment and resistance to ICIs. This complex interaction and balance between the TME and tumor cells triggers immune evasion mechanisms, including T cell exclusion, induction of regulatory T cells (Treg), and other immune suppressor cells, increasing PD-Ll expression, among others. For instance, poor efficacy of ICI monotherapy has been reported in patients with EGFR mutations as a consequence of characteristic low TMB and high expression of PD-L1 in these tumors. Conversely, patients with ALK and ROS1 fusion-positive tumors present a relatively high prevalence of PD-L1 expression, but low TMB and short progression-free survival after monotherapy with ICI, indicating that subsets of NSCLC with EGFR and ALK/ROS1 positive mutations present minimal benefit from ICI despite high PD-L1 expression [206]. In contrast, KRAS mutated NSCLC presents high TMB, increased infiltration of lymphocytes PD-L1+, and an inflammatory tumor microenvironment, being more responsive to ICIs [207]. Therefore, NSCLC cases with distinct genomic subsets and specific oncogenic drivers show heterogeneous response to ICIs. To date, there is limited prospective data on the efficacy of ICIs therapy in NSCLC with driver mutations, mainly because ICIs clinical trials consistently exclude *EGFR*, *ALK,* and *ROS1* mutated tumors, thus precluding meaningful clinical information [208].

As previously acknowledged, patients with active BM are frequently excluded from clinical trials testing ICIs in NSCLC; therefore, the safety and activity of ICIs as a single-agent or combined with chemo or radiotherapy modalities are still under investigation [160,204]. A non-randomized, open-label, phase 2 trial showed that pembrolizumab provides similar response rates in intracranial and extracranial tumors, with improved overall survival in NSCLC BM presenting with PD-L1 expression ≥1% [209]. Therapy with pembrolizumab (anti–PD-1 monoclonal antibody) in patients with treatment-naive and previously treated PD-L1–positive advanced/metastatic NSCLC showed improved outcomes and fewer adverse events compared to chemotherapy alone in a pooled analysis of the Keynote-001, -010, -024, and -042 clinical trials, supporting the use of pembrolizumab monotherapy for these patients [210]. Additionally, Powell et al. [211] reported a pooled analysis of Keynote-021, -189, and -407 including 1298 NSCLC patients, of which 171 had baseline BM. In this study, patients with or without BM, treated with pembrolizumab plus platinum-based chemotherapy, showed improved clinical outcomes vs. chemotherapy alone across all PD-L1 positive samples [211]. In their study, patients with BM treated with pembrolizumab plus chemotherapy had a median overall survival of 18.8 months compared with 7.6 months with chemotherapy alone, and median progression-free survival of 6.9 and 4.1 months, respectively. Therefore, combined treatment regimens were suggested as a standard-of-care option for patients with advanced NSCLC, including those with stable brain metastases [211]. Similarly, the CheckMate 9LA, an international, randomized, open-label, phase 3 trial, showed that treatment with nivolumab plus ipilimumab combined with two cycles of chemotherapy resulted in superior overall survival when compared to chemotherapy alone, and suggested the use of this therapeutic regimen as a first-line option in advanced NSCLC [212]. In a systematic review and meta-analysis, superior overall survival and progression-free survival was reported for patients with advanced NSCLC treated with ICIs compared to chemotherapy alone. This study also reported that a combination of treatment with nivolumab/ipilimumab plus chemotherapy resulted in further improved overall survival and progression-free survival of patients with BM [213].

Although the combination of radiation and targeted therapy or immunotherapy in the management of patients with BM NSCLC is controversial, clinical trials evaluating the role of local radiation with these therapies are ongoing (NCT04905550; NCT02978404; and NCT03916419, among others). Despite encouraging results with systemic therapy, the incidence of BM is still increasing. In addition, CNS progression and therapeutic resistance urgently require combinatorial strategies, including local therapy and novel CNS-penetrant drugs that can adequately treat intracranial metastases.

## 8. Conclusions and Perspectives

Approximately 40% of NSCLC patients develop BM during their disease course, leading to high morbidity and mortality rates. Management of patients with BM is challenging, and a multidisciplinary approach is necessary for treatment and disease control. In light of the increasing incidence of BM and poor clinical management, ongoing advances in multimodal treatments and targeted therapies are needed, including the development of CNS-penetrant agents that adequately target molecular alterations present in BM. In order to achieve effective and personalized treatment approaches for CNS metastases, -omics profiling should be integrated with microenvironment analyses. Research into the genetic variants and ncRNA expression may help stratify the lung cancer population by the risk of developing BM. Identifying the interactions between tumor cells and the brain microenvironment is also a key step in developing treatment strategies to block metastatic progression.

Improved treatment modalities have been implemented with the development of immune checkpoint inhibitors in combination with other systemic therapies. Although there is an observed gain in survival, patients with NSCLC and BM are still underrepresented in clinical trials, and there is a need for an assessment of routine MRI screening and biomarker identification.

The application of screening tools such as an MRI scan to identify patients with a higher risk of developing BM has the potential to improve patient outcomes. This is especially relevant since a proportion of patients with negative neuroimaging screens will develop a BM within one year of diagnosis; however, it remains controversial whether there is a need for routine neuroimage screens in patients with the early clinical stages of NSCLC.

Efforts for functional assessment of metastatic-competent cells have been described with in vitro and in vivo characterization of CTCs in liquid biopsies, as described in this review. Such studies are needed since the molecular mechanisms underlying the metastatic steps are not fully understood, mainly due to most studies focusing on brain lesions only, and not looking into isolating and identifying metastatic-enabling circulating cells. By integrating molecular analyses of BM collected at different time points during tumor evolution, researchers will aid in understanding disease progression in lung and other cancers. Furthermore, the application of advanced technologies, including single cell sequencing, will offer novel opportunities for analysis of CTC-derived metastasis, unlocking key transcriptomic and molecular changes associated with the metastatic cascade. Although high dimensional and more complex single cell sequencing analyses are still challenging, they hold potential for precision oncology in the context of complex and heterogeneous diseases including NSCLC-BM.

Therefore, it is urgent to fully elucidate the molecular mechanisms of BM, aiming at the development of successful therapeutic interventions, which will ultimately change the dismal prognosis of NSCLC patients with BM. Additionally, combined efforts to fully understand disease heterogeneity and metastatic evolution will lead to the development of better diagnostics for early detection of BM before clinical manifestation, improving patient outcomes and providing better chances of cure.

## Figures and Tables

**Figure 1 cancers-15-00722-f001:**
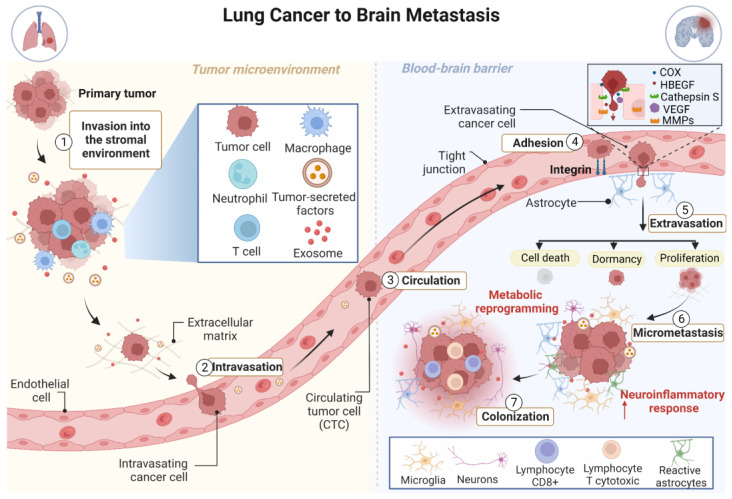
Schematic illustration of brain metastasis development through hematogenous dissemination. Initially, tumor cells at the primary site acquire invasive properties, break away from the primary tumor, and invade the surrounding tissues (intravasation) becoming circulating tumor cells (CTCs). Cell motility is promoted through the interaction between tumor cells and immune cells. Then, CTCs spread throughout the circulatory system to the brain microvasculature (circulation) and after their adhesion with help of integrins, they start the extravasation across the blood–brain barrier (BBB). Tumor cells overexpress enzymes associated with mitogenesis, growth factors, metalloproteinases, and proteases allowing cell migration through the BBB. Once tumor cells are located in the central nervous system (CNS), an intense neuroinflammatory response is triggered. After extravasating, most tumor cells die or enter a state of dormancy (some of them could stay dormant at metastatic sites for long periods). Few tumor cells are able to proliferate within this new microenvironment and then form micrometastases and colonize the brain (colonization). The interaction between tumor cells and immune cells residing in the brain is critical for the establishment and growth of the tumor. COX2: prostaglandin-endoperoxide synthase 2; HBEGF: heparin-binding EGF-like growth factor; MMPs: metalloproteinases; VEGF: vascular endothelial growth factor.

**Figure 2 cancers-15-00722-f002:**
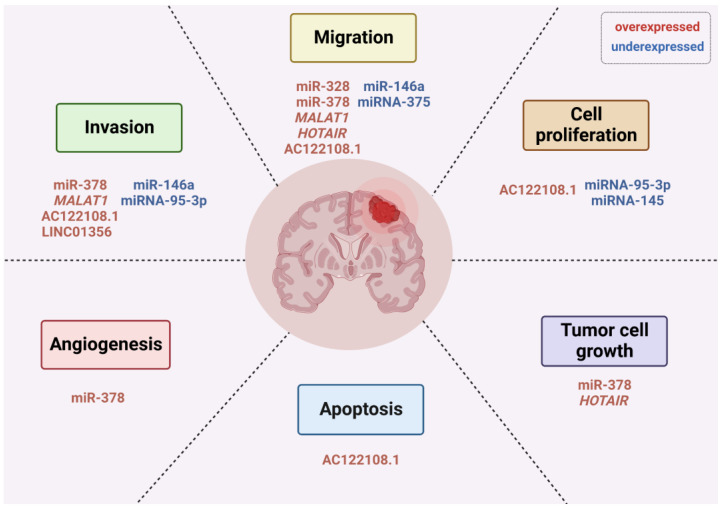
Non-coding RNAs (ncRNAs) and their roles in the hallmarks of cancer implicated in the development of BM from lung cancer. Red color indicates overexpressed ncRNAs; blue color indicates underexpressed ncRNAs.

**Figure 3 cancers-15-00722-f003:**
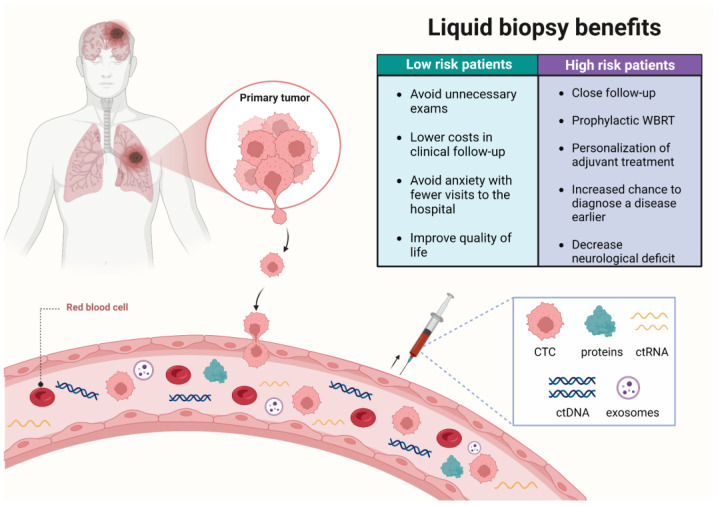
Clinical decisions and benefits for the patients upon using liquid biopsy tests to predict a low or high risk of developing brain metastasis. CTC: circulating tumor cell; ctDNA: circulating tumor DNA; ctRNA: circulating tumor RNA; WBRT: whole brain radiation therapy.

**Figure 4 cancers-15-00722-f004:**
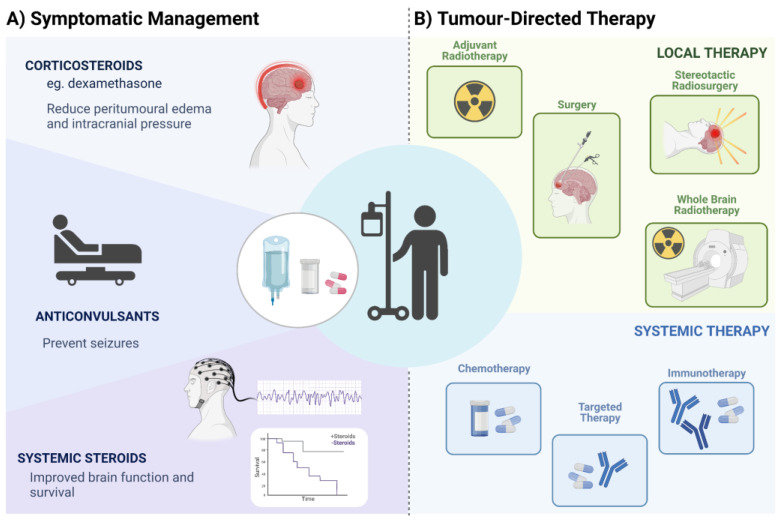
Summary of different strategies for the treatment of patients with brain metastasis including symptomatic management and tumor-directed therapies.

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
