# Peer review of "Advances in the Molecular Landscape of Lung Cancer Brain Metastasis"

_cancers, 2023, doi:10.3390/cancers15030722_

Round 1

Reviewer 1 Report

The review article, “Advances in the molecular landscape of lung cancer brain metastasis” y Souza et al., represents a well-written and thorough examination of the current state of understanding for non small cell lung cancer metastasis to brain. The figures are comprehensive and well-illustrated, and the text is thorough and delves into substantial depth. Overall, there were few substantial concerns and a handful of minor comments noted that could be addressed to further strengthen the review, which are listed in detail below:

Substantial concerns:

·      On page 2, there is a summation of current small molecule inhibitors, all the way down to Alk and ROS. The new Kras inhibitors should also be included.

·      Exosomes and/or LINC01356  should also be mentioned. https://www.ncbi.nlm.nih.gov/pmc/articles/PMC9092220/

·      Remodeling of the blood brain barrier and the role it plays in metastasis permissivity should also be mentioned. https://www.nature.com/articles/s41419-021-04004-z

·      Surgical resection of BMs is minimally discussed on page 11, and there is no mention of the success rates or statistical outcomes.

Minor concerns:

·      Page 2: “However, these therapies benefit a small fraction of patients with lung adenocarcinoma har- boring driver mutations (~15%).”. Depending on the population being studied, EGFR represents a much larger portion of LUAD occurrences than 15% and this is mentioned lower in the paragraph. It would be good to mention these caveats or to soften the statement considering percentages or rework/reorganize the paragraph.

·      The following sentence on page three should include references, “Once trapped, tumor cells overexpress enzymes associated with mitogenesis and growth fac- tors, including prostaglandin-endoperoxide synthase 2 (COX2) and heparin-binding EGF- like growth factor (HBEGF), allowing cell migration through the BBB.”

·      Figure 1 is beautifully illustrated but the font size is tiny and makes it difficult to read.

·      Page 8, the usage of “miR-NAs” is not defined in the sentence, “Taken together, these findings suggest that the loss of tumor suppressor miR- NAs could activate oncogenic pathways that are hallmarks of cancer, contributing to tu- mor development and progression to metastasis.”

·      A lot of emphasis is placed on lncRNAs in lung tumor cells that enhance brain metastasis but little is dedicated to how lncRNAs in brain may contribute to a permissive environment, specifically in astrocytes as mentioned earlier.

·      It’s unclear from the text why “vimentin positive” CTs was emphasized on page 10.

·      Page 10, the inclusion of CA125 calls into question the specificity of these biomarkers for NSCLC as CA125 is a classical biomarker of ovarian cancer. Additional text pertaining to the specificity of these for NSCLC should be included.

·      It should be mentioned that on page 11 when discussing small molecule treatments for brain mets that they fall under the umbrella of pallative care.

Author Response

The review article, “Advances in the molecular landscape of lung cancer brain metastasis” by Souza et al., represents a well-written and thorough examination of the current state of understanding for non small cell lung cancer metastasis to brain. The figures are comprehensive and well-illustrated, and the text is thorough and delves into substantial depth. Overall, there were few substantial concerns and a handful of minor comments noted that could be addressed to further strengthen the review, which are listed in detail below:

Substantial concerns:

  • On page 2, there is a summation of current small molecule inhibitors, all the way down to Alk and ROS. The new Kras inhibitors should also be included.

Answer: In order to address the reviewer's comment, we have included information on the new KRAS inhibitors citing recent literature including a review article that details the developments of KRAS inhibitors to treat advanced lung cancer. Please refer to page 3, highlighted version, as follows:

“Other activating mutations occur in oncogenes such as KRAS and are associated with worse prognosis, with no approved drugs able to efficiently inhibit KRAS activation [5–7] until the recent development of sotorasib, a novel inhibitor shown to efficiently target KRASp.G12C mutation in advanced solid tumors [8]. Another KRAS inhibitor, adagrasib, is under investigation to treat patients with progressive metastatic lung cancer [9]. In a phase 2 clinical trial, sotorasib, which specifically and irreversibly inhibits the KRASp.G12C mutation, was tested in a cohort of 126 NSCLC patients with the majority having previously received systemic platinum-based chemotherapy combined with immunotherapy based on PD-1 or PD-L1 immune checkpoint inhibitors (ICIs). Results showed a complete response in 4/126 patients (3.2%) and a partial response in 42/126 patients (33.9%) with a median duration of response of 11.1 months. These data showed clinical response for relapsed advanced KRAS-mutated NSCLC with disease control obtained in >80% of patients. Median progression-free survival and overall survival were 6.8 months and 12.5 months, respectively [10]. Novel treatments targeting KRAS-mutated tumors are promising; however, patient prognosis remains poor with modest progression-free and overall survival rates, likely due to disease heterogeneity. Recent reviews on targeted therapies including KRAS inhibitors including combination with immunotherapies are available in [11–13].”

  • Exosomes and/or LINC01356  should also be mentioned. https://www.ncbi.nlm.nih.gov/pmc/articles/PMC9092220/

Answer: In order to address this comment, we included a sentence in the main text (page 12, highlighted version), as follows:

“Recent studies have shown that lncRNAs are closely related with the increased permeability of the BBB in BM development and brain tumors [117–120]. In NSCLC, both exosomal-derived LINC01356 and lnc-MMP2-2 were found to increase BBB permeability and promote BM development. While the exosomal lncRNA LINC01356 remodels BBB by targeting cell junction proteins such as Occludin, Claudin, and N-cadherin [121], TGF-β1-mediated exosomal lnc-MMP2-2 may destroy the tight junctions of the BBB hereby facilitating the passage of cancer cells [122,123].”

Furthermore, we changed figure 2 to include the role of lncRNA LINC01356 in promoting cell invasion.

  • Remodeling of the blood brain barrier and the role it plays in metastasis permissivity should also be mentioned. https://www.nature.com/articles/s41419-021-04004-z

Answer: In order to address this comment, we included a sentence in the main text (page 5, highlighted version), as follows:

“Once trapped, tumor cells start the process of crossing the BBB, which is a crucial step in tumor dissemination to the brain. The BBB harbors tight junctions and adherens junctions between the brain endothelial cells, which regulate the flow of ions and nutrients, establishing a selective permeability barrier that protects the brain from blood-derived toxins and restricts the migration of leukocytes and monocytes [37–39]. BBB permeability is highly increased during BM in lung cancer [40], allowing CTCs to penetrate the brain and promote BM development. Several mechanisms associated with BBB remodeling that facilitate the migration of tumor cells through the BBB have been identified including the secretion of various proteases to degrade tight junction components [41–43]."

  • Surgical resection of BMs is minimally discussed on page 11, and there is no mention of the success rates or statistical outcomes.

Answer: As suggested by the reviewer, we added the following paragraph to improve the discussion on surgical resection of brain metastasis (page 17, highlighted version), as follows:

“Overall median survival rates after surgical resection of BM range from 9.8 to 24 months [174–176]. Within one year, survival rates are about 40-45% [174,177]. Factors determining better outcomes and prolonged survival are lower number of BM, lower age, better preoperative clinical performance, absence of extracranial metastasis, and association with other treatment modalities (i.e, Radiotherapy and immunotherapy) [174–179]. For single BM, survival can be longer than two years in cases of complete resection and smaller tumor size [180,181].”

Minor concerns:

  • Page 2: “However, these therapies benefit a small fraction of patients with lung adenocarcinoma harboring driver mutations (~15%).” Depending on the population being studied, EGFR represents a much larger portion of LUAD occurrences than 15% and this is mentioned lower in the paragraph. It would be good to mention these caveats or to soften the statement considering percentages or rework/reorganize the paragraph.

Answer: We reorganized the paragraph (pages 2-3, highlighted version), as follows:

“However, these therapies benefit a fraction of patients with lung adenocarcinoma harboring driver mutations. ”

The following sentence on page three should include references, “Once trapped, tumor cells overexpress enzymes associated with mitogenesis and growth factors, including prostaglandin-endoperoxide synthase 2 (COX2) and heparin-binding EGF- like growth factor (HBEGF), allowing cell migration through the BBB.”

Answer: In order to address the reviewer's comment, we have included reference citations, as indicated (page 4, highlighted version):

“For example, cancer cells overexpress enzymes associated with mitogenesis and growth factors, including prostaglandin-endoperoxide synthase 2 (COX2) and heparin-binding EGF-like growth factor (HBEGF), allowing cell migration through the BBB [44,45].”

Reference 44 - Bos PD, Zhang XH, Nadal C, Shu W, Gomis RR, Nguyen DX, Minn AJ, van de Vijver MJ, Gerald WL, Foekens JA, Massagué J. Genes that mediate breast cancer metastasis to the brain. Nature. 2009 Jun 18;459(7249):1005-9. doi: 10.1038/nature08021. Epub 2009 May 6. PMID: 19421193; PMCID: PMC2698953.

Reference 45 - Cacho-Díaz B, García-Botello DR, Wegman-Ostrosky T, Reyes-Soto G, Ortiz-Sánchez E, Herrera-Montalvo LA. Tumor microenvironment differences between primary tumor and brain metastases. J Transl Med. 2020 Jan 3;18(1):1. doi: 10.1186/s12967-019-02189-8. PMID: 31900168; PMCID: PMC6941297.

  • Figure 1 is beautifully illustrated but the font size is tiny and makes it difficult to read.

Answer: We revised Figure 1 so that it now presents a readable font size.

  • Page 8, the usage of “miRNAs” is not defined in the sentence, “Taken together, these findings suggest that the loss of tumor suppressor miRNAs could activate oncogenic pathways that are hallmarks of cancer, contributing to tumor development and progression to metastasis.”

    Answer: We have rewritten the sentence to clarify the function of the miRNAs in the main text (page 11, highlighted version), as follows:

“Taken together, these findings suggest that the loss of miRNAs with a tumor suppressive role could activate oncogenic pathways that are hallmarks of cancer, contributing to tumor development and progression to metastasis.”

  • A lot of emphasis is placed on lncRNAs in lung tumor cells that enhance brain metastasis but little is dedicated to how lncRNAs in brain may contribute to a permissive environment, specifically in astrocytes as mentioned earlier.

Answer: In order to address the reviewer's point, we have included an explanation on the role of lncRNAs in the brain (page 12, highlighted version), as follows:

“Moreover, lncRNAs may interact with immune cells in the brain and contribute to a permissive environment for tumor growth [124]. For example, glioma cell-derived exosomes are able to transport lncRNA-ATB to astrocytes promoting their activation, which in turn facilitates invasion and migration of glioma cells [125]. In breast cancer cells, loss of lncRNA X-inactive-specific transcript (XIST) triggers the polarization of microglia, resulting in increased expression of cytokines and suppression of T-cell proliferation [126]. Immune suppression is one of the mechanisms by which microglia promotes tumor progression in the brain [126]. Another study showed that JAK2-binding long noncoding RNA can promote breast cancer brain metastasis through a STAT3-dependent mechanism, which mediated recruitment of macrophages into the brain [127]. Also, a recent report demonstrated that lncRNA (BMOR) is important for developing breast-to-brain metastasis by allowing tumor cells to evade immune-mediated killing in the brain microenvironment [128]. Altogether, these studies evidence the importance of lncRNAs for mediating communication between cancer cells and the brain microenvironment.”

  • It’s unclear from the text why “vimentin positive” CTCs was emphasized on page 10.

    Answer: We included vimentin positive CTCs as a means of using molecular diagnostic techniques for detection of circulating tumor cells in blood. In the main text (page 14, highlighted version), we included a brief explanation, as follows:

“Vimentin is a filamentous protein expressed in mesenchymal cells, and it is known to maintain cellular integrity and provide resistance against stress [140]. It has been often recorded in cancers, including NSCLC and BM, as a marker of tumor cell invasion via its expression during aberrant activation of epithelial–mesenchymal transition (EMT) [139]. During EMT, vimentin modulates genes for EMT inducers, as well as some key epigenetic factors. It suppresses cellular differentiation and upregulates their pluripotent potential by inducing genes associated with self-renewability, which increases the stemness of cancer stem cells, facilitating tumor spread and making tumor cells more resistant to treatments [141]. Vimentin overexpression has been associated with increased cancer cell growth, invasion and migration, suggesting its potential application in cancer diagnosis and treatment [140].”

  • Page 10, the inclusion of CA125 calls into question the specificity of these biomarkers for NSCLC as CA125 is a classical biomarker of ovarian cancer. Additional text pertaining to the specificity of these for NSCLC should be included.

    Answer: In order to clarify this issue, we reviewed the text and added information regarding CA125, which has been reported in BM of patients with NSCLC (page 15, highlighted version), as follows:

“Cancer antigen 125 has been used as a clinical tumor marker for prognosis and therapy monitoring in ovarian and breast cancer patients [155]. However, other studies reported CA-125 as a marker for worse prognosis in lung cancer [156] and a prognostic biomarker in BM [154]. Also, high serum levels of lactate dehydrogenase (LDH), CEA, CYFRA 21-1 and CA125 were independently associated with BM in a large cohort of geriatric patients with lung adenocarcinoma [157].”

  • It should be mentioned that on page 11 when discussing small molecule treatments for brain mets that they fall under the umbrella of palliative care.

    Answer: This point was made clear with the inclusion of the sentence “Small molecule treatments have proven beneficial for palliative relief” within the following paragraph (page 17, highlighted version):

“Current guidelines for BM treatment recommend targeted therapy only for those patients with oncogenic driver mutations [158]. Small molecule treatments have proven beneficial for palliative relief. For example, patients with NSCLC BM and positive for EGFR mutation have shown meaningful CNS efficacy after treatment with third generation of EGFR TKIs such as icotinib [185] and osimertinib [186–188]. Similarly, intracranial response was observed in patients with BM treated with third generation of TKIs targeting ALK rearrangements, alectinib [189], brigatinib [190,191], and ceritinib [192]. Other therapies have been explored for NSCLC with BM showing promising results, such as capmatinib [193] and tepotinib [194], both targeting MET alterations; selpercatinib targeting RET fusions [195,196]; entrectinib [197], and larotrectinib targeting tropomyosin receptor kinase (TRK) fusion-positive tumors [198,199]; lorlatinib targeting ROS1 [200]; dabrafenib plus trametinib targeting BRAF V600E [201], among others [161, 202]. ”

Reviewer 2 Report

The review proposed by Vanessa G. P. Souza et al relates to a very important topic which is BMs in NSCLC. However, the current form of the article does not provide sufficient information and I suggest including some major remarks before it is accepted.

1. Immunotherapy plays a crucial role in the treatment of NSCLC, however, little information related to this regimen was indicated in the introduction compared to molecularly targeted therapy.

2. Moreover, the introduction does not present a clear aim of the review.

3. The description of molecular determinates of BM does not present the most up to date up-to-date genetic background of BM or alterations that are associated with BM in NSCLC (PMID: 32203465; PMID: 36561283; PMID: 33987392).

4. The paragraph about the clonality of BM that drives the heterogeneity is very pivotal and should be elaborated in detail as a separate paragraph based on up-to-date original studies on multiregional sequencing of BM or corresponding primary lesions as well available reviews.

5. The liquid biopsy paragraph does not provide sufficient data on how genetic CTC profiling could reveal the detection of BM or how informative could be CTC profiling in CSF.  

6. Conclusions and further perspectives are presented in a too generic way and they are too short in relation to so a long and comprehensive review. I strongly encourage the authors to cross the boundaries of the presented article and provide more info about in vitro/in vivo BM modelling for studies on new molecular targets or the application of single-cell resolution approaches to study the microenvironment or clonal heterogeneity of BM.

Author Response

The review proposed by Vanessa G. P. Souza et al relates to a very important topic which is BMs in NSCLC. However, the current form of the article does not provide sufficient information and I suggest including some major remarks before it is accepted.

  1. Immunotherapy plays a crucial role in the treatment of NSCLC, however, little information related to this regimen was indicated in the introduction compared to molecularly targeted therapy.

Answer: In order to expand the text about immunotherapy in NSCLC, we added more information in the Introduction and Treatment of BM sections, as follows:

Introduction section (pages 3-4):

“Conversely, patients with squamous cell lung carcinoma, which represents about 20%-30% of NSCLC, have limited treatment options. Treatment with biomarker-driven therapies targeting FGFR, PI3K, MET, EGFR, among others, failed to demonstrate activity in the Lung Cancer Master Protocol (Lung-MAP SWOG S1400). However, an ongoing phase 2 open label clinical trial (RAGNAR) showed evidence of efficacy for erdafitinib, a selective pan-FGFR tyrosine kinase inhibitor, in heavily pretreated patients with different FGFR-positive solid tumors, including squamous and non-squamous cell lung cancer [14]. In the last decade, immunotherapy based on immune checkpoint inhibitors (ICIs) has shown significant survival benefits for patients with advanced NSCLC. Cancer cells develop immune evasion mechanisms playing a pivotal role in cancer progression. Monoclonal antibodies, such as pembrolizumab and nivolumab, are directed to block PD-1 receptor in T lymphocytes, preventing immune response inhibition [1,15]. Patients with advanced NSCLC treated with ICIs have improved survival in comparison to standard chemotherapy in both first- and second-line therapies. Efficacy of nivolumab monotherapy in pretreated advanced non-squamous and squamous cell lung cancer showed a 17% objective rate response (ORR) and median of 17.0 months of response duration among patients [16]. Combination of different ICIs with distinct and complementary mechanisms to improve anti-tumor immunity, such as nivolumab targeting PD-1 and ipilimumab targeting CTLA-4 in T lymphocytes, was tested in a phase 1, multi-cohort study showing high response rate and durable response with tolerable safety in NSCLC [17] .”

Treatment of BM from lung cancer section (pages 17-18):

“Immunotherapy using ICIs are now commonly used in the management of NSCLC, particularly for patients without molecularly targetable disease. The benefit of ICIs in  oncogene-targetable NSCLC is limited. Cumulative evidence suggests an interplay with  tumor cell oncogenic signaling and tumor immunogenicity, leading to non-T cell-inflamed environment and resistance to ICIs. This complex interaction and balance between the TME and tumor cells triggers immune evasion mechanisms, including T cell exclusion,  induction of regulatory T cells (Treg) and other immune suppressor cells, increasing PD-Ll expression, among others. For instance, poor efficacy of ICI monotherapy has been reported in patients with EGFR mutations as a consequence of characteristic low TMB and high expression of PD-L1 in these tumors. Conversely, patients with ALK and ROS1 fusion-positive tumors present relatively high prevalence of PD-L1 expression, but low TMB and short progression-free survival after monotherapy with ICI, indicating that subsets of NSCLC with EGFR and ALK/ROS1 positive mutations present minimal benefit from ICI despite high PD-L1 expression [207]. In contrast, KRAS mutated NSCLC presents high TMB, increased infiltration of lymphocytes PD-L1+ and an inflammatory tumor microenvironment, being more responsive to ICIs [208]. Therefore, NSCLC cases with distinct genomic subsets and specific oncogenic drivers show heterogeneous response to ICIs. To date, there is limited prospective data on the efficacy to ICIs therapy in NSCLC with driver mutations, mainly because ICIs clinical trials consistently exclude EGFR, ALK and ROS1 mutated tumors, thus precluding meaningful clinical information [209].”

  1. Moreover, the introduction does not present a clear aim of the review.

Answer: In order to make the review´s aim clear, we have modified the abstract text, and included a paragraph at the end of Introduction (page 4, highlighted version), as follows:

“The goal of this review is to summarize the current state of knowledge on the mechanisms of metastatic spread of lung cancer to the brain; how metastatic spread is influenced by the brain microenvironment, and to elucidate the molecular determinants of brain metastasis regarding the role of genomic and transcriptomic changes including coding and non-coding RNAs. We also present an overview of the current therapeutics and novel treatment strategies for patients diagnosed with BM from NSCLC.“

“In this Review, we describe the current state of knowledge regarding the molecular and cellular mechanisms involved in metastatic spreading of lung cancer cells to the brain. We discuss the influence of the brain microenvironment including immune cells to support tumor cell growth. Moreover, a comprehensive discussion of genomic and transcriptomic alterations, including coding and non-coding RNAs, as genetic determinants of brain metastasis in NSCLC is presented. We also provided an overview of the current therapeutics, new treatment opportunities, and future directions for patients diagnosed with BM from NSCLC.”

  1. The description of molecular determinates of BM does not present the most up to date up-to-date genetic background of BM or alterations that are associated with BM in NSCLC (PMID: 32203465; PMID: 36561283; PMID: 33987392).

Answer: In order to address this important point raised by the reviewer, we have included a separate paragraph with relevant information from the published studies, as suggested (page 7, highlighted version):

“Also, genetic alterations driving BM formation/progression were previously reported. Whole-exome sequencing of 73 BM cases from lung adenocarcinoma (BM-LUAD) identified MYC, YAP1, MMP13 amplifications and CDKN2A/B deletions as pathogenic genomic changes [68]. Additionally, it was demonstrated that overexpression of these candidate driver genes (MYC, YAP1, or MMP13) promoted BM in patient-derived xenograft (PDX) mouse models [68]. In another study, by comparing focal somatic copy number alterations (SCNAs) in matched NSCLC-BM pairs, putative BM-driving genetic alterations were identified affecting multiple cancer genes, including several potentially targetable changes in genes such as CDK12, DDR2, ERBB2, and NTRK1 [69]; these results were validated in an independent cohort of 84 BM samples and characterized SCNAs and mutations in EP300, CTCF, and STAG2 genes, which play roles in epigenome editing and 3D genome organization [69]. Whole exome sequencing analysis of 12 paired primary NSCLC and matched BM have also identified BM-associated mutations in known cancer genes including AHNAK2, ANKRD36C, BAGE2, KMT2C, and PDE4DIP [70].“

  1. The paragraph about the clonality of BM that drives the heterogeneity is very pivotal and should be elaborated in detail as a separate paragraph based on up-to-date original studies on multiregional sequencing of BM or corresponding primary lesions as well available reviews.

Answer: In order to address this important point raised by the reviewer, we have included separate paragraphs, as suggested, indicating relevant and up-to-date literature on clonality and heterogeneity in brain metastasis. Please see pages 8-9, highlighted version, as follows:

“BMs may harbor high genetic heterogeneity and clonal differences between their corresponding primary tumors, suggesting that additional molecular changes may be acquired during metastatic progression [67]. A number of studies have been performed in an attempt to address the question of clonality and molecular heterogeneity between primary tumors and brain metastasis from same patients. Some studies have collected and profiled metastatic lesions in an asynchronous mode with the primary tumor, allowing detection of evolutionary changes over time. In a report by Lee et al [71], multi-omics sequencing of seven paired tumors and brain metastasis (collected at different time points) from patients with NSCLC, showed that 67% of mutations were common between metastatic and primary samples. In addition, these lesions had similar tumor mutational burden (TMB). Further validation using publicly available data from a whole exome sequencing study of 35 brain metastases and primary samples [72] showed 69% of shared mutations and similar TMB frequency. Based on these findings, the authors suggested that metastatic events occur late during the evolutionary tumor development and progression cycles, likely upon the establishment of the majority of somatic mutations in the primary tumor [71]. Although the results of this study are based on a small sample set of 7 patients, the authors also suggested that a monoclonal mode of metastatic seeding may be predominant in most NSCLC cases.

Interestingly, Brastianos et al. [72] identified that, although there are genetic similarities between BM lesions arising in different brain sites as well as temporally separated, there is high genetic heterogeneity between BM and lymph node metastasis or extracranial distant metastasis. In addition, they reported actionable changes in BM, correlated with drug sensitivity to PI3K/AKT/mTOR, CDK, and HER2/EGFR inhibitors [72]. Other studies found similar results [73] and reported molecular changes likely selected during metastatic progression, such as deletions of CDKN2A/B which were common to metastatic and primary samples [68]. A recent whole exome sequencing study of 84 tissue samples from 26 patients compared genomic profiles of primary lung adenocarcinoma, liver and brain metastasis lesions; this study showed common driver mutations in TP53 and EGFR in primary and metastatic samples. Additionally, a comparable TMB was present in all samples, however the liver metastases had higher TMB and were more similar to the primary tumors than the BM lesions [74]. These authors also performed phylogenetic analyses and found that liver metastasis were genetically divergent from the paired primary tumors at a later stage of metastatic development compared to BM sites, suggesting that liver metastasis may arise preferably through a linear mode and BM may be established following a parallel mode of progression. It is important to highlight some differences among published studies, which may be due to different patient cohorts, distinct methodologies of sample collection with metastatic samples being collected either synchronously or asynchronously with the primary tumors, and different platforms of analyses. Although the current knowledge on the genetic divergence and phylogenetic evolutionary relationships among BM lesions and primary tumors, this is still an area that deserves further and detailed investigation.”

  1. The liquid biopsy paragraph does not provide sufficient data on how genetic CTC profiling could reveal the detection of BM or how informative could be CTC profiling in CSF. 

Answer: We have added up-to-date information and data to improve the information regarding CTC detection in BM as well as CTC profiling in CSF (page 14, highlighted version), as follows:

“CTCs detected by liquid biopsies will help understand the molecular aspects of metastatic progression. Indeed, CTCs have been indicated as clinically applicable for many years and a plethora of studies have demonstrated the correlation between CTC counts and metastatic disease in different cancer types [145]. In NSCLC patients with BM, CSF has been shown to be useful for CTC detection [146,147]. Recent functional studies have established in vitro and in vivo models from CTC-derived metastatic cells, and are valuable to reveal molecular alterations specific to aggressive, metastatic-enabling clones [144]. Further development of methods for detection of metastasis-initiating CTCs will help elucidate the processes by which metastasis is established into the brain and extracranial sites. Darlix et al [148]  reported a prospective study for detection of suspected leptomeningeal metastasis in 40 patients with breast cancer. The authors tested the CellSearch® system, a clinically validated and FDA-approved test for CTC detection [148], and were able to identify CTCs in cerebrospinal fluid of all cytology-positive samples. Interestingly, they detected CTCs in five cytology-negative samples, demonstrating improved sensitivity of CTC detection using the CellSearch® system. Furthermore, they were able to detect HER-2 positive CTCs in CSF of HER-2 negative tumors. This same system has been previously used to evaluate detection of both CTCs and exosomes in pancreatic cancer patients, and was shown useful to accelerate diagnosis and treatment of surgically resectable cases [149]. Such studies highlight the importance of liquid biopsies as a potential tool to study molecular changes specific to more aggressive circulating tumor cells, as well as to refine molecular diagnostics and aid in treatment decisions that will impact patient survival [148]. Furthermore, the development of liquid biopsy-CTC-based biomarkers can be useful as a complementary tool to aid diagnostic imaging, augmenting early detection and, consequently, treatment intervention of occult BM [150].”

  1. Conclusions and further perspectives are presented in a too generic way and they are too short in relation to a long and comprehensive review. I strongly encourage the authors to cross the boundaries of the presented article and provide more info about in vitro/in vivo BM modeling for studies on new molecular targets or the application of single-cell resolution approaches to study the microenvironment or clonal heterogeneity of BM.

Answer: In order to address the reviewer´s suggestions, we have included more information on advanced studies that have been performed to characterize metastatic-potent circulating cells in in vitro and in vivo models. We also included the application of single-cell resolution approaches to better understand the complexity of the brain metastatic microenvironment, as promising future directions for early detection and effective treatment of brain metastasis. This is included on pages 19-20, as follows:

“Efforts for functional assessment of metastatic-competent cells have been described with in vitro and in vivo characterization of CTCs in liquid biopsies, as described in this review. Such studies are needed since the molecular mechanisms underlying the metastatic steps are not fully understood, mainly due to most studies focusing on brain lesions only, and not looking into isolating and identifying metastatic-enabling circulating cells. By integrating molecular analyses of BM collected at different time points during tumor evolution, researchers will aid in understanding disease progression in lung and other cancers. Furthermore, the application of advanced technologies, including single cell sequencing, will offer novel opportunities for analysis of CTC-derived metastasis, unlocking key transcriptomic and molecular changes associated with the metastatic cascade. Although high dimensional and more complex single cell sequencing analyses are still challenging, they hold potential for precision oncology in the context of complex and heterogeneous diseases including NSCLC-BM.

Therefore, it is urgent to fully elucidate the molecular mechanisms of BM, aiming at the development of successful therapeutic interventions, which will ultimately change the dismal prognosis of NSCLC patients with BM. Additionally, combined efforts to fully understand disease heterogeneity and metastatic evolution will lead to the development of better diagnostics for early detection of BM before clinical manifestation, improving patient outcomes and providing better chances of cure.”

In addition, we added new data from the literature to highlight the role of single cell sequencing analysis and other advanced technologies of spatial RNA profiling in NSCLC-BM (page 10, highlighted manuscript version), as follows:

“Recent studies applying single cell transcriptomic sequencing were able to unlock novel key molecular features of individual metastatic cells in lung cancer. Among these, Ruan et al [80] characterized transcriptome changes in circulating tumor cells in cerebrospinal fluid of patients with lung adenocarcinoma leptomeningeal metastasis. They sequenced 792 transcriptomes of 5 patients and 3 controls, and found a metastatic signature of genes with roles in metabolism as well as molecules related to cell adhesion. Furthermore, this study reported that there is higher heterogeneity among patients when compared to single cells isolated from the same patient [80].  In addition, Zhang et al [81], identified differentially expressed genes between primary lung adenocarcinoma and BM isolated from two patient derived xenografts (PDX). The authors suggested that CKAP4 (Cytoskeleton Associated Protein 4), SERPINA1 (Serpin Family A Member 1), SDC2 (Syndecan 2) and GNG11 (G Protein Subunit Gamma 11) are potential biomarkers to aid in prognosis assessment and therapy of patients with lung cancer BM [81].

Other technologies were applied for digital spatial RNA sequencing profiling of NSCLC and BM, and allowed to comprehensively assess biomarkers associated with primary and metastatic lesions including analysis of the primary tumor immune and the brain microenvironments. Zhang et al [82] analyzed a cohort of 44 patients with metastatic NSCLC using the NanoString GeoMx DSP platform. Among their findings, we highlight the highly immunosuppressive microenvironment associated with BM lesions compared to primary tumors, with reduced abundance of B and T-cells and higher infiltration of neutrophils. Their study shed light into the role of molecular changes in the tumor and BM microenvironments for establishment of the metastatic niche [82]. Studies such as these are relevant to determine clinically applicable biomarkers for patient treatment.”

Reviewer 3 Report

1. Suggest including comments regarding lung cancer screening. Is lung cancer screening increasing and will it have an impact on lung cancer stage patterns and mortality? If more early stage lung cancers are found, could this scenario have implications for identifying tumor molecular profiles associated with high risk of developing brain metastases? 

2.  While KRAS inhibitors have been approved for treatment of lung cancer patients whose tumors harbor KRAS mutations, these treatments modest response rates and are associated with modest improvement in progression free and overall survival. Suggest including comment about these observations. 

3. In your targeted therapies section, it would be good to include more recent studies: Shaw AT 2019 lorlatinib fro ROS 1 rearragements, Dziadziusko R 2021 entrectinib for ROS1 rearrangements & is effective in brain metastases, Blanchard D 2016 dabrafenib and tremetinib for BRAF mutations, Hong DS 2020 larotrectonib for NTRK fusions, Doebele RC 2020 entrectinib for NTRK fusions. 

4. In your discussion about immunotherapy regimens, it would be good to include references describing relatively high response rates in brain metastases and superior overall survival in patients treated with immunotherapy/chemotherapy combinations. Powell SF 2021 J Thorac Oncol, Carbone D 2021 J Thorac Oncol 16, S862. 

5. On page, do 47% of lung cancer patients have brain metastases at time of initial diagnosis? 

6. In your comments regarding side effects of glucocorticoids, it would be good to include steroid induced myopathy and osteoporosis with increased risk of fractures. 

7. In your comments regarding effectiveness of targeted therapies for brain metastases, it would be good to include Shaw AT 2019 efficacy of lorlatinib for ROS1 positive brain metastases, Dziadiusko R 2021 efficacy of entrectinig for ROS1 positive brain metastases. 

8. It would be good to consider the following comments and questions in your conclusion:

a. Will brain MRI brain scans increase frequency of detecting brain metastases?

b. Will screening increase the number of early stage lung cancer patients and are continued efforts to identify molecular profiles associated with increased risk of subsequent brain metastases warranted? 

c. Are continued efforts to identify targeted therapies that are effective in brain metastases with driver mutations ongoing? 

d. Do preliminary results with immunotherapy for brain metastases warrant continued study in larger groups of patients with brain metastases?  

Author Response

  1. Suggest including comments regarding lung cancer screening. Is lung cancer screening increasing and will it have an impact on lung cancer stage patterns and mortality? If more early stage lung cancers are found, could this scenario have implications for identifying tumor molecular profiles associated with high risk of developing brain metastases? 

Answer: In order to address the points raised by the reviewer, we included the following information (page 4, highlighted version):

“Routine brain MRI would increase the detection of asymptomatic brain metastasis. However, its use as a populational guideline is controversial due to the high burden on the patients and the health care system [25,26]. Besides, a proportion of patients with negative screens may develop brain metastasis within one year [27]. Therefore, current guidelines support routine neuroimaging scans for more advanced clinical stages.”

Additionally, we addressed the need for identification of molecular profiles and biomarkers associated with BM in the following paragraph (please see page 10, highlighted version):

“Considering the reported evolutionary changes and genetic differences between primary tumors and BM, as well as the molecular features that result from selective pressures of systemic treatments, there is a need for additional molecular profiling studies leading to biomarker development in BM. The relevance of novel biomarker discovery in BM and also extracranial metastasis include the identification of molecular determinants/drivers of metastatic progression, and the application of this knowledge for more effective treatment approaches to improve patient survival [79].”

  1. While KRAS inhibitors have been approved for treatment of lung cancer patients whose tumors harbor KRAS mutations, these treatments modest response rates and are associated with modest improvement in progression free and overall survival. Suggest including comment about these observations. 

Answer: We addressed the reviewer´s comment by including results from a clinical trial with Sotorasib, which has been tested for NSCLC patients with KRASp.G12C mutation. Please find these data on Page 3, highlighted version:

“In a phase 2 clinical trial, sotorasib, which specifically and irreversibly inhibits the KRASp.G12C mutation, was tested in a cohort of 126 NSCLC patients with the majority having previously received systemic platinum-based chemotherapy combined with immunotherapy based on PD-1 or PD-L1 immune checkpoint inhibitors (ICIs). Results showed a complete response in 4/126 patients (3.2%) and a partial response in 42/126 patients (33.9%) with a median duration of response of 11.1 months. These data showed clinical response for relapsed advanced KRAS-mutated NSCLC with disease control obtained in >80% of patients. Median progression-free survival and overall survival were 6.8 months and 12.5 months, respectively [10]. Novel treatments targeting KRAS-mutated tumors are promising; however, patient prognosis remains poor with modest progression-free and overall survival rates, likely due to disease heterogeneity. Recent reviews on targeted therapies including KRAS inhibitors including combination with immunotherapies are available in [11–13].”

  1. In your targeted therapies section, it would be good to include more recent studies: Shaw AT 2019 lorlatinib for ROS 1 rearragements, Dziadziusko R 2021 entrectinib for ROS1 rearrangements & is effective in brain metastases, Blanchard D 2016 dabrafenib and tremetinib for BRAF mutations, Hong DS 2020 larotrectonib for NTRK fusions, Doebele RC 2020 entrectinib for NTRK fusions. 

Answer: We have added up-to-date information and included the recommended references (page 17, highlighted version), as follows:

“Current guidelines for BM treatment recommend targeted therapy only for those patients with oncogenic driver mutations [158]. Small molecule treatments have proven beneficial for palliative relief. For example, patients with NSCLC BM and positive for EGFR mutation have shown meaningful CNS efficacy after treatment with third generation of EGFR TKIs such as icotinib [185] and osimertinib [186–188]. Similarly, intracranial response was observed in patients with BM treated with third generation of TKIs targeting ALK rearrangements, alectinib [189], brigatinib [190,191], and ceritinib [192]. Other therapies have been explored for NSCLC with BM showing promising results, such as capmatinib [193] and tepotinib [194], both targeting MET alterations; selpercatinib targeting RET fusions [195,196]; entrectinib [197], and larotrectinib targeting tropomyosin receptor kinase (TRK) fusion-positive tumors [198,199]; lorlatinib targeting ROS1 [200]; dabrafenib plus trametinib targeting BRAF V600E [201], among others [161,202].

  1. In your discussion about immunotherapy regimens, it would be good to include references describing relatively high response rates in brain metastases and superior overall survival in patients treated with immunotherapy/chemotherapy combinations. Powell SF 2021 J Thorac Oncol, Carbone D 2021 J Thorac Oncol 16, S862. 

Answer: We have added further information to improve our discussion on the benefits of combined chemotherapy and immunotherapy treatment regimens to treat patients with brain metastasis, as suggested by the reviewer (please see page 18, highlighted version):

“Additionally, Powell et al [212] reported a pooled analysis of Keynote-021, -189, and -407 including 1298 NSCLC patients, of which 171 had baseline BM. In this study, patients with or without BM, treated with pembrolizumab plus platinum-based chemotherapy, showed improved clinical outcomes vs. chemotherapy alone across all PD-L1 positive samples [212]. In their study, patients with BM treated with pembrolizumab plus chemotherapy had median overall survival of 18.8 months compared with 7.6 months with chemotherapy alone, and median progression-free survival of 6.9 and 4.1 months, respectively. Therefore, combined treatment regimens were suggested as a standard-of-care option for patients with advanced NSCLC, including those with stable brain metastases [212]. Similarly, the CheckMate 9LA, an international, randomized, open-label, phase 3 trial, showed that treatment with nivolumab plus ipilimumab combined with two cycles of chemotherapy resulted in superior overall survival when compared to chemotherapy alone, and suggested the use of this therapeutic regimen as a first-line option in advanced NSCLC [213]. In a systematic review and meta-analysis, superior overall survival and progression-free survival was reported for patients with advanced NSCLC treated with ICIs compared to chemotherapy alone. This study also reported that combination of treatment with nivolumab/ipilimumab plus chemotherapy resulted in further improved overall survival and progression-free survival of patients with BM [214].”

  1. On page, do 47% of lung cancer patients have brain metastases at time of initial diagnosis? 

Answer: We have reviewed this information, as pointed by the reviewer, and reorganized the paragraph (page 4, highlighted version), as follows:

“Approximately 10-20% of NSCLC patients have BM at the time of diagnosis and approximately 40% will develop BM during the course of disease  [21,22].”

  1. In your comments regarding side effects of glucocorticoids, it would be good to include steroid induced myopathy and osteoporosis with increased risk of fractures. 

Answer: We have added the side effects information (please see page 16, highlighted version), as follows:

“Despite the advantages of corticosteroid use, their symptom relief is temporary and dependent on the control of the local disease. Besides, the side effects of prolonged use are well known, such as diabetes, weight gain, cushingoid features, hypertension, myopathy, and osteoporosis - with increased risk of fractures [168].”

  1. In your comments regarding effectiveness of targeted therapies for brain metastases, it would be good to include Shaw AT 2019 efficacy of lorlatinib for ROS1 positive brain metastases, Dziadiusko R 2021 efficacy of entrectinig for ROS1 positive brain metastases. 

Answer: In order to address the reviewer´s comment, we have included information regarding the effectiveness of targeted therapies for brain metastases citing the recommended papers. The information is found in the manuscript section: Treatment of BM from lung cancer, page 17, highlighted version, as follows:

“The reduced penetration of chemotherapy agents through BBB has limited its use as primary treatment for BM in NSCLC [145]. Pemetrexed-cisplatin was shown to be effective for treatment of BM in patients with NSCLC with an objective response rate of 41.9% [183]. The FDA-approved drug Entrectinib was developed to target NTRK fusion as well as ROS and ALK tyrosine kinases and is capable of crossing the BBB. This drug has shown positive intracranial response rates in ROS positive NSCLC [203,204], and NTRK fusion-positive solid tumors [197]. Also, Lorlatinib, a potent brain-penetrant, third-generation tyrosine kinase inhibitor, has shown clinical activity in patients with advanced ROS1-positive NSCLC with BM [200]. Moreover, chemotherapy combined with immunotherapy has been shown to enhance the efficacy of immunotherapy, opening new windows for new first-line therapeutic strategies to benefit patients with advanced NSCLC [205,206].”

  1. It would be good to consider the following comments and questions in your conclusion:
    1. Will brain MRI brain scans increase frequency of detecting brain metastases?
    2. Will screening increase the number of early stage lung cancer patients and are continued efforts to identify molecular profiles associated with increased risk of subsequent brain metastases warranted?
    3. Are continued efforts to identify targeted therapies that are effective in brain metastases with driver mutations ongoing? 
    4. Do preliminary results with immunotherapy for brain metastases warrant continued study in larger groups of patients with brain metastases?

Answer: These points have been addressed in the conclusion section (pages 19-20), as follows:

“Improved treatment modalities have been implemented with the development of immune checkpoint inhibitors in combination with other systemic therapies. Although there is an observed gain in survival, patients with NSCLC and BM are still underrepresented in clinical trials, and there is a need for assessment of routine MRI screening and biomarker identification.

The application of screening tools such as MRI scan to identify patients with a higher risk of developing BM has the potential to improve patient outcomes. This is specially relevant since a proportion of patients with negative neuroimaging screens will develop a BM within one year of diagnosis; however, it remains controversial whether there is a need for routine neuroimage screens in patients with early clinical stages of NSCLC.

Efforts for functional assessment of metastatic-competent cells have been described with in vitro and in vivo characterization of CTCs in liquid biopsies, as described in this review. Such studies are needed since the molecular mechanisms underlying the metastatic steps are not fully understood, mainly due to most studies focusing on brain lesions only, and not looking into isolating and identifying metastatic-enabling circulating cells. By integrating molecular analyses of BM collected at different time points during tumor evolution, researchers will aid in understanding disease progression in lung and other cancers. Furthermore, the application of advanced technologies, including single cell sequencing, will offer novel opportunities for analysis of CTC-derived metastasis, unlocking key transcriptomic and molecular changes associated with the metastatic cascade. Although high dimensional and more complex single cell sequencing analyses are still challenging, they hold potential for precision oncology in the context of complex and heterogeneous diseases including NSCLC-BM.

Therefore, it is urgent to fully elucidate the molecular mechanisms of BM, aiming at the development of successful therapeutic interventions, which will ultimately change the dismal prognosis of NSCLC patients with BM. Additionally, combined efforts to fully understand disease heterogeneity and metastatic evolution will lead to the development of better diagnostics for early detection of BM before clinical manifestation, improving patient outcomes and providing better chances of cure.”